# Neuroinflammation-induced lymphangiogenesis near the cribriform plate contributes to drainage of CNS-derived antigens and immune cells

Martin Hsu[1], Aditya Rayasam[1], Julie A. Kijak[2], Yun Hwa Choi[3], Jeffrey S. Harding[4], Sarah A. Marcus [2], William J. Karpus[2], Matyas Sandor[2] & Zsuzsanna Fabry[2]

There are no conventional lymphatic vessels within the CNS parenchyma, although it has been hypothesized that lymphatics near the cribriform plate or dura maintain fluid homeostasis and immune surveillance during steady-state conditions. However, the role of these lymphatic vessels during neuroinflammation is not well understood. We report that lymphatic vessels near the cribriform plate undergo lymphangiogenesis in a VEGFC – VEGFR3 dependent manner during experimental autoimmune encephalomyelitis (EAE) and drain both CSF and cells that were once in the CNS parenchyma. Lymphangiogenesis also contributes to the drainage of CNS derived antigens that leads to antigen specific T cell proliferation in the draining lymph nodes during EAE. In contrast, meningeal lymphatics do not undergo lymphangiogenesis during EAE, suggesting heterogeneity in CNS lymphatics. We conclude that increased lymphangiogenesis near the cribriform plate can contribute to the management of neuroinflammation-induced fluid accumulation and immune surveillance.

---

[1] Neuroscience Training Program, University of Wisconsin-Madison, Madison, WI 53705, USA. [2] Department of Pathology and Laboratory Medicine, University of Wisconsin-Madison, Madison, WI 53705, USA. [3] School of Pharmacy, University of Wisconsin-Madison, Madison, WI 53705, USA. [4] Samuel Lunenfeld Research Institute, Mount Sinai Hospital, Toronto, ON M5T 3L9, Canada. These authors contributed equally: Matyas Sandor, Zsuzsanna Fabry. Correspondence and requests for materials should be addressed to Z.F. (email: zfabry@wisc.edu)

Lymphatic vessels regulate cell trafficking, antigen drainage, and fluid homeostasis within tissues of the body[1,2]. Lymphatic vessels typically reside within the tissue parenchyma and facilitate drainage of fluid and antigens to the draining lymph nodes. Recently, lymphatic vessels surrounding the central nervous system (CNS) have been re-characterized under steady-state conditions, yet it is unclear how antigens or immune cells from the CNS parenchyma migrate into lymphatics in the dura or cribriform plate during neuroinflammation[3–5]. Alternative routes of drainage for CSF or immune cells from the CNS have also been proposed: (1) along olfactory cranial nerves penetrating the cribriform plate, (2) along other cranial nerves such as the optic nerve, (3) through arachnoid villi into the venous sinuses, and (4) within perivascular spaces, or the "glymphatic" system[5–10]. The relative contribution(s) of each pathway to the drainage of CSF, lymphocytes, and antigens during neuroinflammation are controversial[11–15]. Improper drainage of CSF may lead to edema and limit the drainage of antigens. Understanding the regulatory mechanisms of CNS drainage is critical for understanding how neuroinflammation is managed.

Lymphangiogenesis is critical during development, systemic inflammation, wound healing, tumor spread, and immunity[1]. During development, lymphatic endothelial cells proliferate and undergo Vascular Endothelial Growth Factor Receptor 3 (VEGFR3)-dependent lymphangiogenesis in the meninges[16,17]. In adulthood, meningeal lymphatics can still undergo lymphangiogenesis; injection of the VEGFR3 ligand recombinant VEGFC or AAV-mVEGFC into the cisterna magna induces lymphatic vessel widening in the superior sagittal sinus[3,17]. However, adult lymphangiogenesis has not been well characterized in lymphatics surrounding the CNS during neuroinflammation. Nevertheless, lymphangiogenesis in peripheral organs is associated with several pathologies including tissue transplant rejection[18–21] and is important for managing inflammation, edema, and T cell responses[22–24]. Since the expression of several members of the VEGF family are up-regulated within the CNS and correlate with disease severity in multiple sclerosis (MS) and in experimental autoimmune encephalomyelitis (EAE)[25,26], we hypothesize that EAE-induced neuroinflammation may promote lymphangiogenesis surrounding the inflamed CNS.

To investigate the drainage of dendritic cells from the CNS during neuroinflammation, we induced EAE in CD11c-eYFP transgenic reporter mice and observed lymphangiogenesis near the cribriform plate 18 days post-immunization. We focused on lymphangiogenesis near the cribriform plate and on their functionality, mechanism, and contribution to CNS autoimmunity during EAE. We show that EAE induces VEGFR3-dependent lymphangiogenesis, which can carry cells that were once in the CNS parenchyma, CD11c-eYFP+ cells, and CSF. CCL21 is also up-regulated within the CNS during EAE, and correlates with increased CCR7+ CD11c-eYFP+ cell accumulation within lymphangiogenic vessels near the cribriform plate. Inhibition of VEGFR3 reduces the drainage of CNS-derived antigens to the draining lymph nodes, reduces EAE severity, and correlates with reduced CD4 T cell infiltration and demyelination in the spinal cord. Our data suggest that neuroinflammation can recruit dendritic cells and monocytes to induce VEGFR3-dependent lymphangiogenesis and identify VEGFR3 as a novel player in the initiation of EAE.

## Results

### Characterization of lymphatics near the cribriform plate.
It has been demonstrated that CSF can be collected by the cribriform plate lymphatics or nasal lymphatics[7,8]. However, the precise anatomical location of lymphatic vessels near the cribriform plate has not been well defined, and it is uncertain whether lymphatic vessels in the nasal mucosa are able to penetrate through the cribriform plate and connect to lymphatics on the CNS side[8,27]. In order to visualize the precise anatomical location of lymphatic vessels and their relation to the cribriform plate, we prepared whole-head coronal sections after decalcification for immunohistochemistry (Fig. 1a; Supplementary Fig. 1). We employed the lymphatic endothelial cell transgenic reporter Prox1-tdTomato mouse to visualize lymphatic vessels[28]. Whole-head coronal sections of healthy Prox1-tdTomato transgenic mice were immunolabeled with Lyve-1, a hyaluronan receptor primarily expressed by lymphatic vessels, and vessels near the cribriform plate were positive for both of these markers (Fig. 1b–d). We also observed non-cellular unspecific labeling of Prox1/Lyve-1 near the outer layers of the olfactory bulbs, potentially due to autofluorescence[29]. Additionally, these vessels were positive for VEGFR3, a tyrosine kinase receptor that contributes to lymphangiogenesis in the presence of its ligand VEGFC[30] (Fig. 1e–g) as well as Podoplanin, a glycoprotein thought to play a role in linking the vascular and lymphatic systems[31] (Fig. 1h–j). CCL21, a chemokine implicated in the migration of immune cells into lymphatic vessels[32–35], is also expressed by these vessels (Supplementary Fig. 2). These lymphatic vessels are found on the CNS side of the cribriform plate and penetrate through the cribriform plate to the non-CNS side (Fig. 1; Supplementary Fig. 3). These data confirm previous observations[17,27].

### EAE induces lymphangiogenesis near the cribriform plate.
Lymphatic vessels near the cribriform plate are hypothesized to drain antigen-presenting cells such as dendritic cells during neuroinflammation. To visualize antigen-presenting cell drainage during neuroinflammation, we induced EAE in CD11c-eYFP transgenic reporter mice and immunolabeled whole-head coronal sections for lymphatic vessels with Lyve-1 at day 18 post-immunization. We observed significant lymphangiogenesis near the cribriform plate during EAE compared to healthy controls. Nine serial coronal sections spanning 1920 μm revealed larger Lyve-1+ vessel area in EAE mice across all sections (Fig. 2a, b). Analysis of the section with the largest Lyve-1+ vessel area (Fig. 2a, red inset, Fig. 2b, yellow inset) shows larger Lyve-1+ vessel area during EAE compared to controls (Fig. 2c–h). Quantitation of Lyve-1+ vessel area and volume across all sections confirmed our observations of lymphangiogenesis during EAE near the cribriform plate (Fig. 2i, j). We observed both the expansion and widening of pre-existing lymphatic vessels dorsally and laterally at the CNS side of the cribriform plate. Higher magnification or 3D reconstruction of these lymphatic vessels can be seen in Supplementary Fig. 3 and Supplementary Movies 1 and 2.

Lymphangiogenesis can occur through proliferation of pre-existing lymphatic endothelial cells[21,36–38], or potentially through trans-differentiation of infiltrating monocytes into lymphatic endothelial cells[39–41]. To visualize if lymphangiogenesis can at least be partially attributed to proliferation of pre-existing lymphatic endothelial cells, we immunolabeled healthy and EAE sections for the proliferation marker Ki67 (Fig. 2k–t; Supplementary Figs. 4–6; Supplementary Movies 3–6). Quantitation of the average percent of Ki67+ nuclei within Lyve-1+ vessels revealed a slight but significant increase in lymphatic endothelial cell proliferation in EAE mice compared to healthy controls (Fig. 2u). We cannot exclude the possibility of trans-differentiation by infiltrating monocytes as a mechanism of lymphangiogenesis at this time, and future experiments are needed to address this. Consequently, neuroinflammation-induced lymphangiogenesis near the cribriform plate is due to

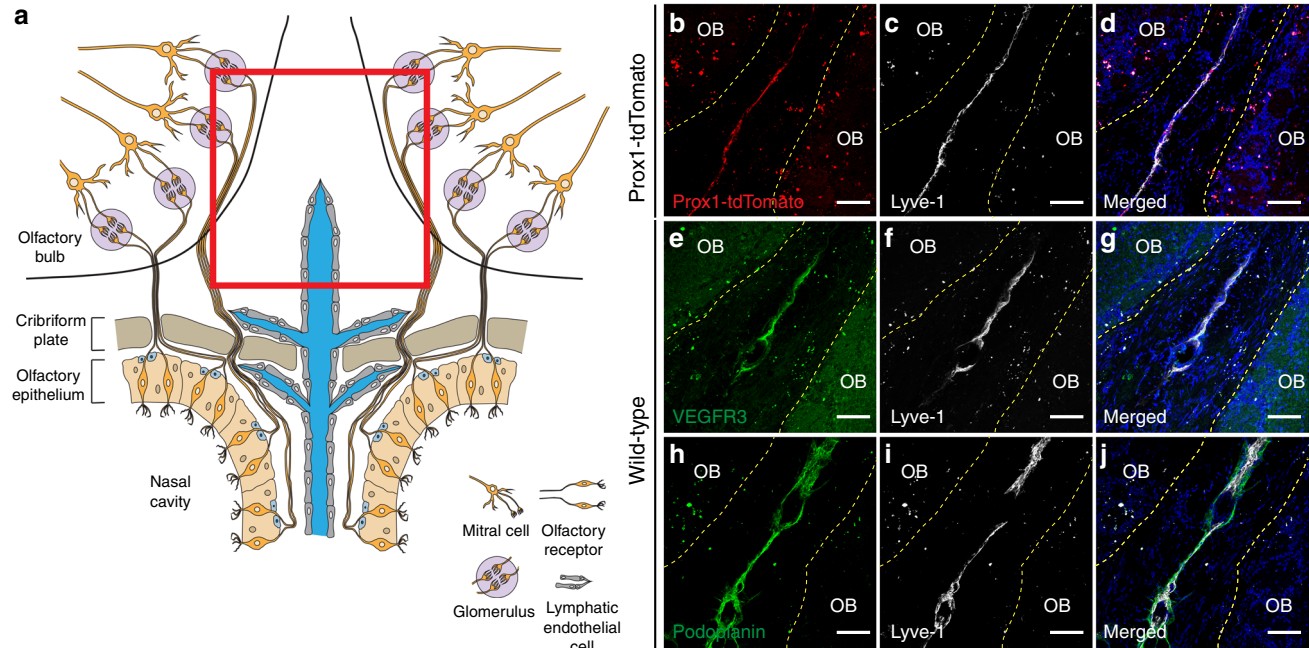

**Fig. 1** Cribriform plate lymphatics exist dorsal to the cribriform plate. **a** Diagram of a coronal section through the olfactory bulbs, cribriform plate, and olfactory epithelium of a mouse. The red inset indicates the region of vessels characterized in **b–j**. Note that these vessels penetrate from the nasal septum through the cribriform plate and into the subarachnoid space. **b–d** The whole heads of healthy naive Prox1-tdTomato transgenic mice were decalcified, coronally sectioned, and immunolabeled with the lymphatic hyaluronan receptor Lyve-1 (**c**) and DAPI (**d**). Scale bars = 100 μm. **e–j** The whole heads of healthy naive wild-type mice were decalcified, coronally sectioned, and immunolabeled with the lymphangiogenic tyrosine kinase receptor VEGFR3, Lyve-1, and DAPI (**e–g**) or the lymphatic marker Podoplanin, Lyve-1, and DAPI (**h–j**). Scale bars = 100 μm

the proliferation of pre-existing lymphatic endothelial cells with trans-differentiation of infiltrating monocytes as a potential alternative mechanism.

**Lymphangiogenic vessels drain cells and CSF**. Next, we assessed whether lymphangiogenic vessels near the cribriform plate are able to functionally drain immune cells and CSF-injected tracers during EAE[42]. To test whether lymphangiogenic vessels can carry dendritic cells, we induced EAE in CD11c-eYFP transgenic reporter mice and confirmed elevated levels of CD11c-eYFP+ cells inside and outside of the CNS parenchyma compared to healthy controls[43] (Fig. 3a–s). We quantified CD11c-eYFP+ specifically within lymphatic vessels, excluding any resident CNS cell, and observed a significant increase in CD11c-eYFP cell number in EAE (Fig. 3i; Supplementary Fig. 5). These data suggest that lymphangiogenic vessels at the cribriform plate carry CD11c-eYFP+ cells during EAE.

Several groups have hypothesized that CNS-derived immune cells migrate towards the cribriform plate during EAE, although this has never been conclusively shown[43–45]. In order to address whether cells from the CNS parenchyma can migrate into cribriform plate lymphatics during neuroinflammation, we induced EAE in KikGR transgenic mice which constitutively express the Kikume Green-Red photoconvertible protein[46] (Supplementary Fig. 1). During EAE, we intracerebrally photoconverted cells within the CNS parenchyma using a 405 nm light coupled fiber optic cable funneled through an 18-gauge needle. Photoconversion of cells within the brain was confirmed by confocal microscopy (Fig. 3t–w) and allowed us to track CNS parenchyma-derived cells draining to the cribriform plate lymphatics and consequently to the cervical lymph nodes during neuroinflammation. The trauma caused by intracerebrally funneling the fiber optic cable on two consecutive days may recruit additional immune cells for photoconversion.

Photoconverted cells were detected in both cribriform plate lymphatics (Fig. 3x–A_b) and the draining lymph nodes (Fig. 3B_b–E_b). Images of photoconverted cells within cribriform plate lymphatics represent approximately 3–8 cells per animal. These data implicate cribriform plate lymphatic vessels as a potential route for the drainage of cells from the CNS parenchyma. While we cannot determine the identity of these cells without additional immunolabeling, photoconverted cells found outside of the CNS parenchyma and within cribriform plate lymphatics exclude CNS-resident cells.

Lymphangiogenic vessels also contribute to the management of fluid homeostasis during inflammation. In order to see if EAE-induced lymphangiogenic vessels drain CSF, we injected 10% Evans blue into the cisterna magna of EAE mice (Supplementary Fig. 1). Whole-head coronal sections immunolabeled with Lyve-1 reveal Evans blue co-localizing with lymphatic vessels near the cribriform plate in EAE mice, suggesting that these lymphangiogenic vessels are able to drain CSF during EAE (Fig. 3F_b–J_b). Additionally, magnetic resonance imaging confirmed previous reports of CSF-infused tracer accumulation near the base of the brain, the cribriform plate, and the deep cervical lymph nodes over time[47] (Supplementary Fig. 6). These data suggest that lymphangiogenic vessels near the cribriform plate can carry both cells and CSF.

**EAE-induced lymphangiogenesis requires VEGFR3**. Lymphangiogenesis occurring in the meninges during development or in peripheral organs after infection have been reported to depend on VEGFR3 and its ligand VEGFC[16,17,21]. Injection of recombinant human VEGFC into the cisterna magna of naïve adult mice increased lymphatic vessel diameter in the superior sagittal sinus[3]. To determine if lymphangiogenesis near the cribriform plate during EAE is dependent on VEGFR3, we treated wild-type mice with the VEGFR3 tyrosine kinase inhibitor MAZ51 once

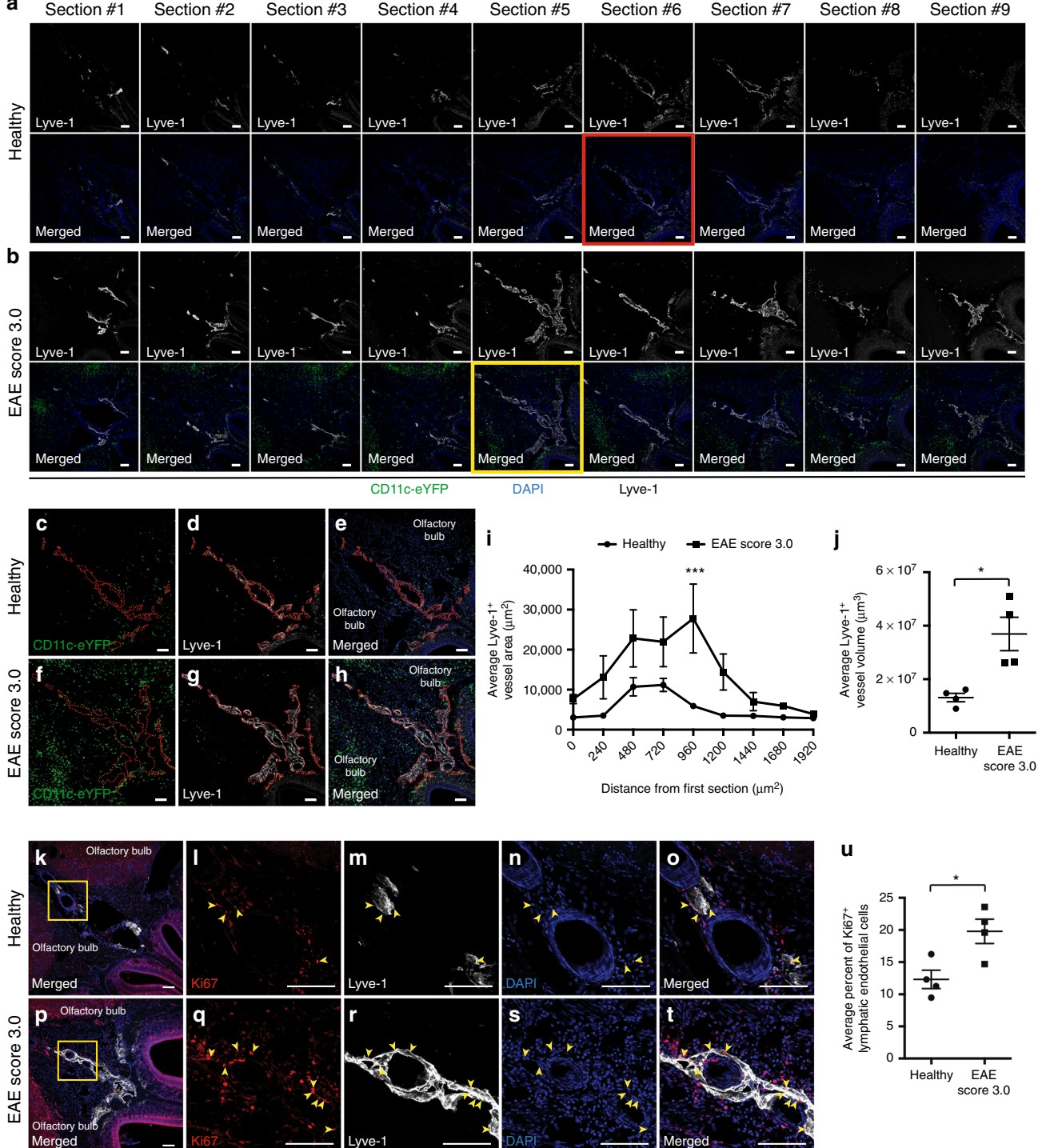

**Fig. 2** EAE induces lymphangiogenesis near the cribriform plate. **a**, **b** Nine representative serial coronal sections of the lymphatic vessels near the cribriform plate were imaged from either a healthy (**a**) or an EAE Score 3.0 collected 18 days post-immunization (**b**) CD11c-eYFP transgenic reporter mouse were immunolabeled with Lyve-1 and DAPI. Note the increase in Lyve-1+ vessel area during EAE versus healthy mice. Scale bars = 100 μm. **c–h** Enlarged images of the red inset in **a** or yellow inset in **b** showing the section with the largest Lyve-1+ vessel area in healthy (**c–e**) versus EAE Score 3.0 (**f–h**) out of nine serial coronal sections. Scale bars = 100 μm. **i**, **j** Quantitation of the average Lyve-1+ vessel area across nine serial sections (**i**) ($n = 4$ mice per group; data are represented as mean ± SEM, $***p < 0.001$, repeated two-way ANOVA using Sidaks multiple comparison test) or the average Lyve-1+ vessel volume (**j**) ($n = 4$ mice per group; data are represented as mean ± SEM, $*p < 0.05$, unpaired Student's $t$-test). **k–t** Proliferation of lymphatic endothelial cells measured by Ki67+ nuclei within healthy (**k–o**) or EAE Score 3.0 mice (**p–t**). Ki67+ Lyve-1+ lymphatic endothelial cells were quantified using orthogonal views to confirm that the cells used for quantitation were in fact proliferating lymphatic endothelial cells. Yellow arrowheads indicate Ki67+ nuclei within Lyve-1+ vessels. Scale bars, 100 μm in **a–t**. Scale bars = 100 μm. **u** Quantitation of the average percent of Ki67+ nuclei within Lyve-1+ vessels between healthy and EAE Score 3.0 mice ($n = 4$ mice per group; data are represented as mean ± SEM, $*p < 0.05$, unpaired Student's $t$-test

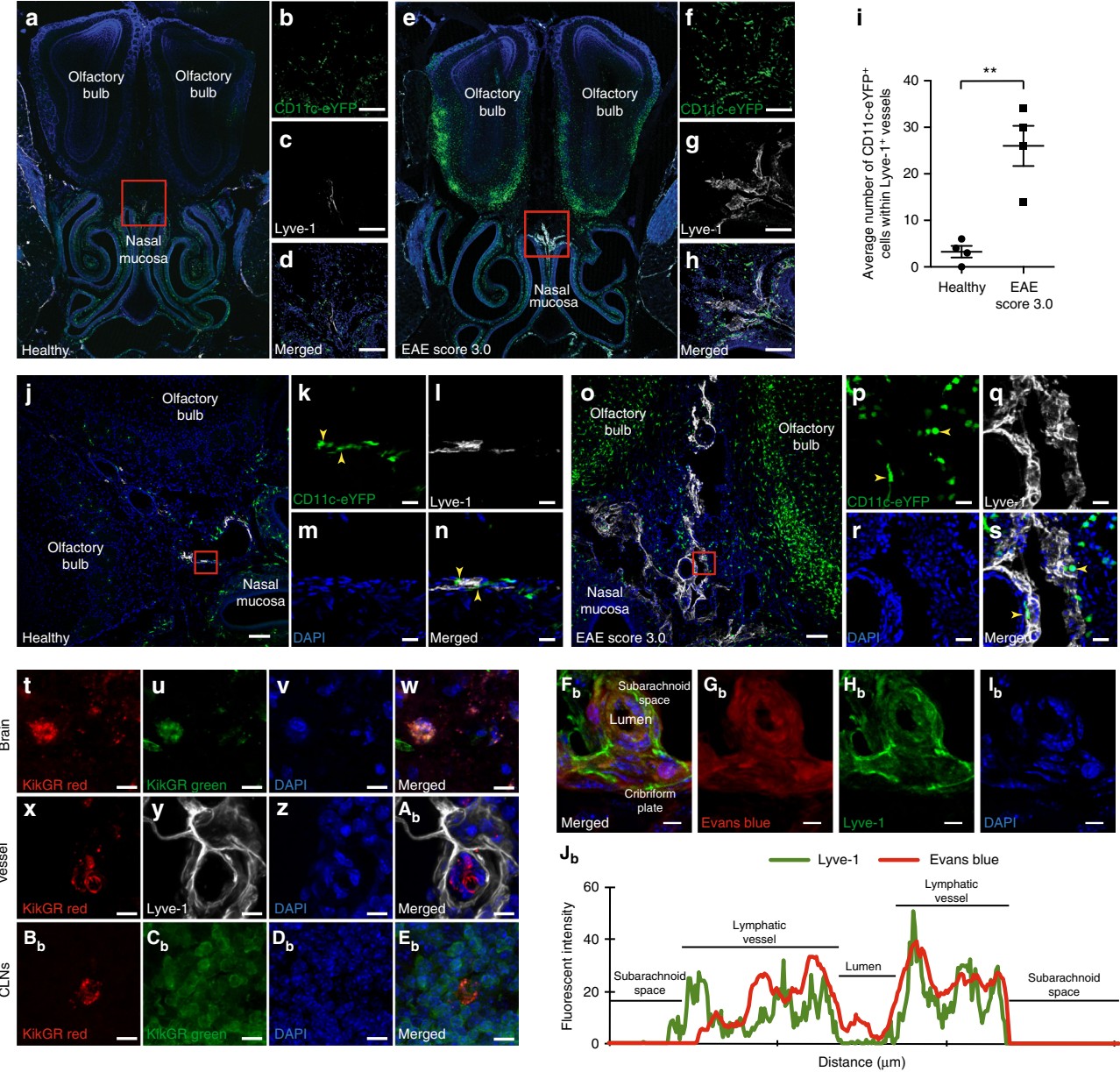

**Fig. 3** Lymphangiogenic vessels can functionally carry cells and CSF during EAE. **a–h** Representative confocal images of the olfactory bulbs, cribriform plate, and nasal mucosa were imaged from a healthy (**a–d**) or EAE Score 3.0 (**e–h**) and immunolabeled with Lyve-1 and DAPI. **b–d** Magnified image of the red inset shown in **a**. **f–h** Magnified image of the red inset shown in **e**. Scale bars = 100 μm. **i** Quantitation of CD11c-eYFP+ cell number within Lyve-1+ vessels. Orthogonal views were used for quantitation to confirm that the CD11c-eYFP+ cells quantified were within Lyve-1+ vessels (n = 4 mice per group; data are represented as mean ± SEM, **p < 0.01, unpaired Student's t-test). **j, s** Higher magnification images from either healthy (**j–n**) or EAE Score 3.0 (**o–s**) CD11c-eYFP transgenic reporter mice showing CD11c-eYFP+ cells within Lyve-1+ vessels. **k–n** Magnified image of the red inset in **j**. **p, s** Magnified image of the red inset in **o**. Scale bars = 100 μm for (**j, o**) and 20 μm for (**k–n, p–s**). **t–E_b** KikGR transgenic mice were induced with EAE, and at EAE score 3.0 were intracerebrally photoconverted to visualize if photoconverted cells originating from the CNS parenchyma can be found within lymphangiogenic vessels near the cribriform plate and the draining lymph nodes. Representative images show green-red photoconverted cells at the site of photoconversion within the brain (**t–w**), within lymphatic vessels near the cribriform plate (**x–A_b**), and within the cervical lymph nodes (CLNs) (**B_b–E_b**). **F_b–I_b**: Wild-type animals were induced with EAE, and at peak EAE 10 μl of 10% Evans blue dye was injected into the cisterna magna at a rate of 2 μl/min to visualize the drainage of CSF within cribriform plate lymphatics. Representative images show co-localization (**F_b**) of Evans blue dye (**G_b**) within Lyve-1+ vessels (**H_b**) and DAPI (**I_b**). Scale bars = 10 μm. **J_b** Representative intensity profile plot for Evans blue dye and Lyve-1 taken from a cross section of the image shown in **F_b** showing co-localization of Evans blue dye within Lyve-1+ lymphatic endothelial cells

per day, beginning on day 7 post-immunization to prevent any unspecific effects of the drug in the initial priming and expansion of T cells. MAZ51 has been shown to be specific for VEGFR3 with only partial inhibition of VEGFR1 and VEGFR2 at higher concentrations[48]. MAZ51 has previously been optimized to

inhibit lymphangiogenesis in vivo without affecting antigen-specific T cell proliferation in vitro[18].

Vehicle-treated EAE mice exhibited extensive lymphangiogenesis at the cribriform plate, consistent with our previous observations in Fig. 2 (Fig. 4a, b). MAZ51 had no effect on

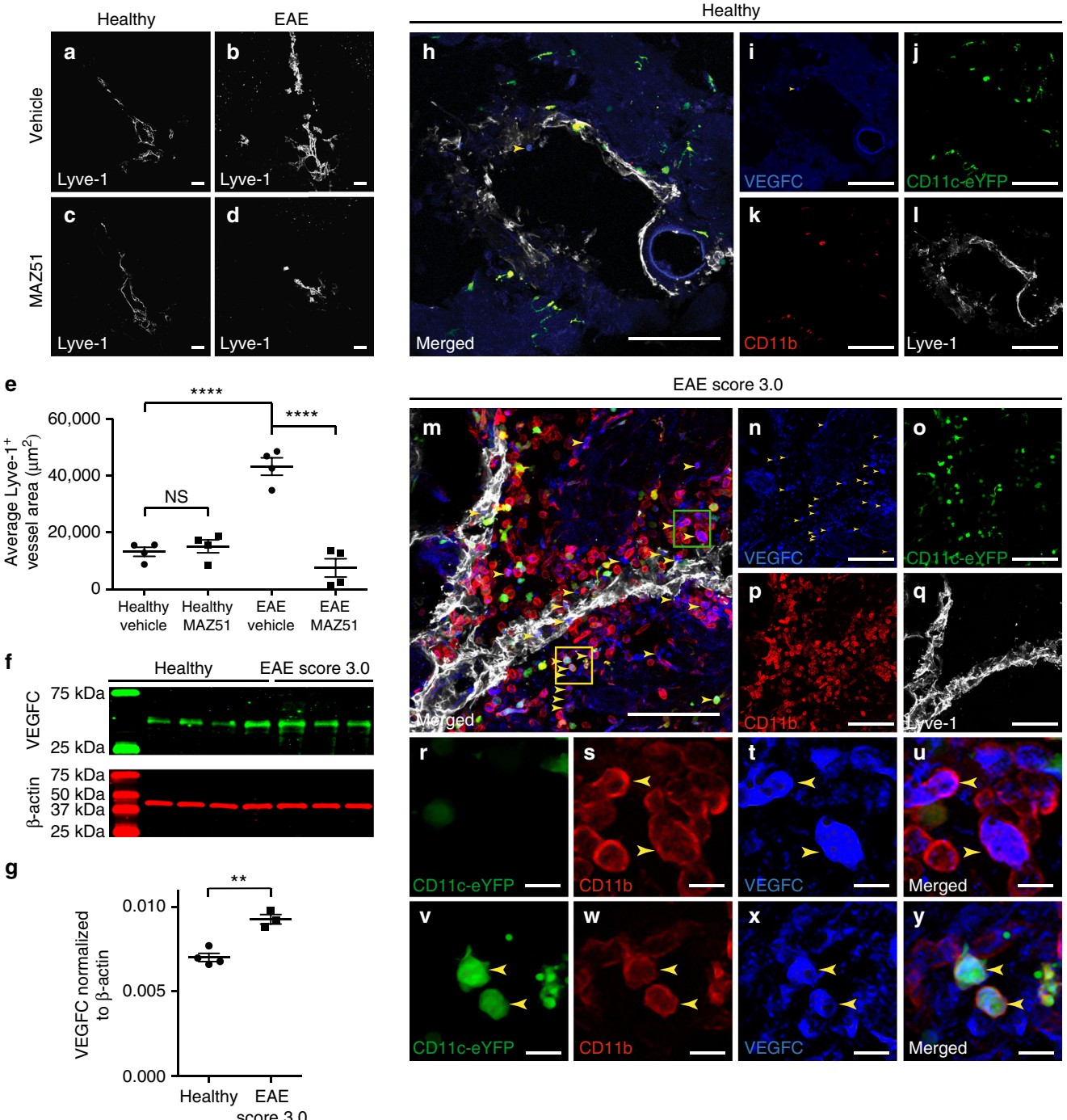

**Fig. 4** EAE increases VEGFC to promote VEGFR3-dependent lymphangiogenesis. **a–d** Healthy (**a**, **c**) and EAE (**b**, **d**) wild-type mice were treated I.P. with either vehicle (**a**, **b**) or the VEGFR3 tyrosine kinase inhibitor MAZ51 (**c**, **d**) beginning on day 7 post-immunization and harvested at Day 18 post-immunization. The whole heads were decalcified, and coronal sections were immunolabeled for Lyve-1 to visualize lymphatic vessels near the cribriform plate. Scale bars = 100 μm. **e** Quantitation of the average Lyve-1+ vessel area near the cribriform plate between vehicle and MAZ51-treated mice in both healthy and EAE. For each animal, nine serial sections spanning 1920 μm were analyzed and the section with the maximum Lyve-1+ vessel area was used for quantitation (mean ± SEM; n = 4 mice per group; ****p < 0.0001, one-way ANOVA with Tukeys post-hoc multiple comparisons test). **f** CNS lysates from healthy and EAE score 3.0 wild-type mice were probed for VEGFC and ß-actin as a loading control by western blot. **g** Quantitation of the relative VEGFC protein amount normalized to ß-actin as shown in **d** (mean ± SEM; n = 3–4 mice per group; **p < 0.01, unpaired Student's t-test). **h–q** Representative coronal section of a healthy (**h–l**) or EAE Score 3.0 (**m–q**) CD11c-eYFP transgenic reporter mice immunolabeled for CD11b, VEGFC, Lyve-1, and DAPI to visualize VEGFC-producing cells near cribriform plate lymphatics. Scale bars = 100 μm. **r–u** Magnified image of the green inset taken from **m** showing co-localization of CD11c-eYFP− CD11b+ macrophages with VEGFC. Yellow arrowheads indicate CD11c-eYFP− CD11b+ macrophages that express VEGFC. **v–y** Magnified image of the yellow inset taken from **m** showing CD11c-eYFP+ CD11b+ dendritic cells expressing VEGFC. Yellow arrowheads indicate CD11c-eYFP+ CD11b+ dendritic cells that express VEGFC. Scale bars = 10 μm

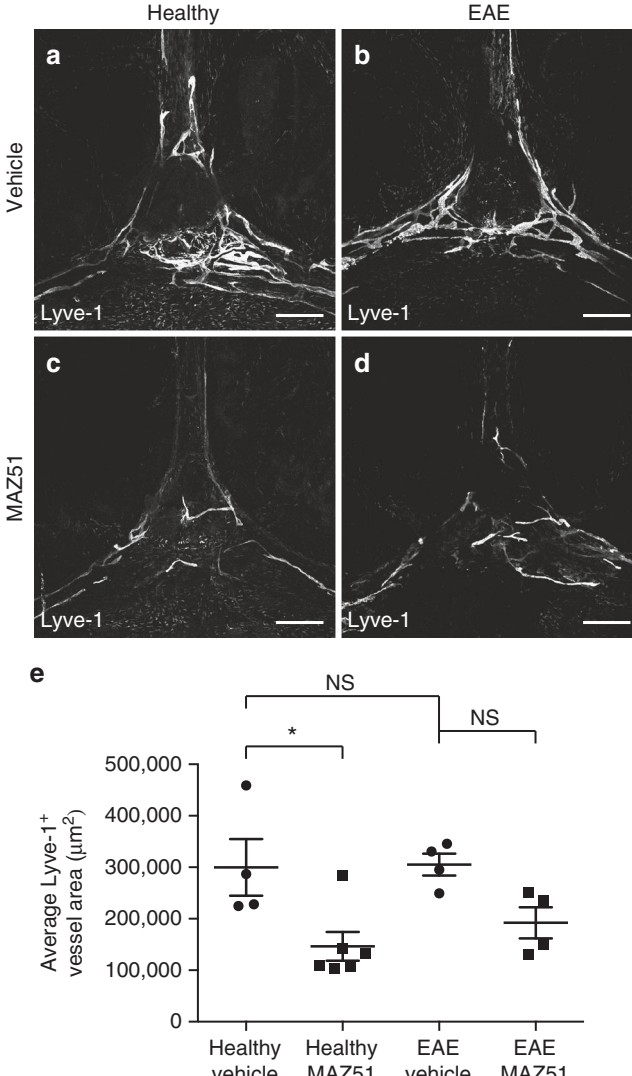

**Fig. 5** Lymphangiogenesis is unique to the cribriform plate during EAE. **a–d** Healthy (**a**, **c**) and EAE Score 3.0 (**b**, **d**) wild-type mice were treated I.P. with either vehicle (**a**, **b**) or the VEGFR3 tyrosine kinase inhibitor MAZ51 (**c**, **d**) beginning on day 7 post-immunization and harvested at day 18 post-immunization. The meninges were collected and immunolabeled for Lyve-1 to visualize meningeal lymphatics. Scale bars = 500 μm. **e** Quantitation of the average Lyve-1+ vessel area in the confluence of sinuses of the dura between vehicle and MAZ51 treated mice in both healthy and EAE (mean ± SEM; $n = 4$–6 mice per group; *$p < 0.05$, one-way ANOVA with Tukeys post hoc multiple comparisons test)

baseline levels of cribriform plate lymphatics in healthy mice (Fig. 4c). In contrast, EAE mice receiving MAZ51 had significantly reduced Lyve-1+ vessel area, suggesting that VEGFR3 is required for lymphangiogenesis in this model (Fig. 4d). Quantitation of average Lyve-1+ vessel area confirmed our observation of reduced lymphangiogenesis after MAZ51 treatment in EAE (Fig. 4e). The VEGFR3 ligand VEGFC is also up-regulated within the CNS during EAE[49] as we confirmed by western blot (Fig. 4d, e). To identify the cellular source(s) of VEGFC, we immunolabeled healthy and EAE CD11c-eYFP transgenic reporter mice with CD11b, VEGFC, and DAPI. While healthy CD11c-eYFP transgenic mice have relatively sparse numbers of CD11c-eYFP cells, CD11b+ cells, and/or VEGFC+ cells near the cribriform plate (Fig. 4f–j), EAE CD11c-eYFP transgenic mice have elevated numbers of CD11b+ VEGFC+ cells

(Fig. 4k, green inset, p–s) and CD11c-eYFP+ CD11b+ VEGFC+ cells near the cribriform plate (Fig. 4t, yellow inset, t–w). While the majority of VEGFC-producing cells are CD11b+ monocytes, several CD11b− cells also express VEGFC, suggesting that there are potentially other sources of VEGFC near the cribriform plate during neuroinflammation. These data implicate infiltrating CD11b+ macrophages and CD11c+ CD11b+ dendritic cells as cellular sources of VEGFC to promote VEGFR3-dependent lymphangiogenesis during neuroinflammation. These cells show similarity to previously described VEGFC-producing cells in peripheral organs during systemic inflammation[41]. These data suggest that recruited dendritic cells and monocytes may serve as critical regulators of CNS drainage through the regulation of lymphangiogenesis.

**Lymphangiogenesis is unique to the cribriform plate**. Recently, dural lymphatic vessels have been characterized and implicated in regulating neuroinflammation[32]. Additionally, meningeal lymphatics have the potential to undergo VEGFR3-dependent lymphangiogenesis after infusion of VEGFC into the CSF[3]. VEGFC is up-regulated in the CNS during EAE[42] (Fig. 4), and here we show that macrophages/dendritic cells produce VEGFC near lymphatics dorsal to the cribriform plate. Therefore, we hypothesize that meningeal lymphatics may also undergo lymphangiogenesis during EAE. Surprisingly, meningeal lymphatics within the confluence of sinuses between the transverse and superior sagittal sinus show no evidence of lymphangiogenesis during EAE (Fig. 5a, b), which has also been confirmed by others[32]. Despite the lack of lymphangiogenesis, meningeal lymphatics remained plastic as they underwent regression after VEGFR3 inhibition as previously described[17] (Fig. 5a–e). These data suggest that lymphoid vessels in the dura and cribriform plate are different, and their relative access to pro-lymphangiogenic factors differ during neuroinflammation. Additionally, MAZ51 treatment during EAE also induces meningeal lymphatic regression although to a somewhat lesser extent when compared to steady-state conditions, suggesting that during neuroinflammation meningeal lymphatics may be slightly less sensitive to VEGFR3 tyrosine kinase inhibitors (Fig. 5e). Additionally, no changes to meningeal lymphatics surrounding the spinal cord were observed[17,32] (Supplementary Fig. 7).

**EAE correlates with CCR7+ CD11c-eYFP+ cell accumulation**. Next, we looked to identify a potential mechanism of how CD11c+ cells migrate from the CNS parenchyma into lymphatic vessels near the cribriform plate. We have previously shown that the migratory chemokine receptor CCR7 is required for dendritic cell emigration from the CNS to the draining lymph nodes during EAE[32,50], similar to how T lymphocytes exit peripheral tissues[51]. CCL21, a ligand for CCR7, is expressed by lymphatics and is hypothesized to drive the chemotaxis of immune cells into lymphatics[33–35]. Several groups have hypothesized that dendritic cells infiltrating the CNS migrate towards the ventral regions of the olfactory bulbs near the cribriform plate during EAE, suggesting that lymphatic vessels near the cribriform plate may facilitate the drainage of dendritic cells[43–45]. Here, we show that CCL21 is expressed by lymphatic vessels dorsal to the cribriform plate during steady-state conditions; however, this is accompanied by relatively sparse numbers of CCR7+ CD11c-eYFP+ cells (Supplementary Fig. 2). During EAE, CCL21-expressing lymphangiogenic vessels near the cribriform plate are accompanied by increased numbers of CCR7+ CD11c-eYFP+ cells (Supplementary Fig. 2). Both CCR7 and CCL21 are up-regulated within the CNS during EAE (Supplementary Fig. 2). These data correlate

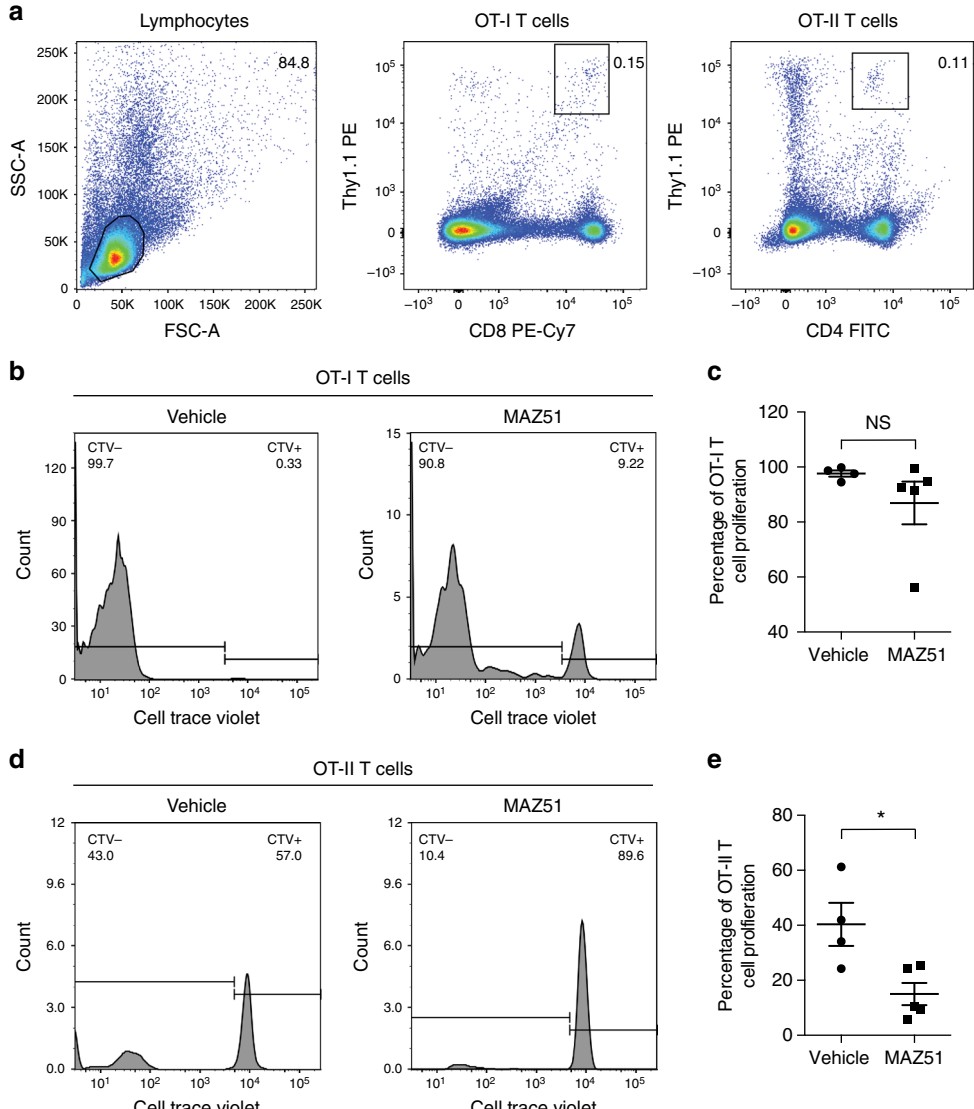

**Fig. 6** Inhibition of VEGFR3 reduces CNS antigen-specific CD4 T cell proliferation. **a–e** ß-gal floxed ovalbumin peptides (OP) transgenic mice were crossed with CNP-Cre transgenic mice to drive ovalbumin peptide expression on oligodendrocytes (CNP-OP mice). These mice were then induced with EAE, I.P. treated with either vehicle or the VEGFR3 tyrosine kinase inhibitor MAZ51 beginning on day 7 post-immunization, and ovalbumin-specific CD8 (OT-I) or CD4 (OT-II) Thy1.1 congenic CellTrace Violet labeled T cells were adoptively transferred on day 12 post-immunization. T cell proliferation was then measured by cytofluorimetry. **a** Gating strategy for the adoptively transferred, CellTrace Violet labeled, Thy1.1 congenic, Ovalbumin-specific T cells. **b** Representative histograms of CellTrace Violet proliferation for OT-I T cells taken from the draining lymph nodes between vehicle and MAZ51 treated EAE animals. **c** Quantitation of the average percent of OT-I T cell proliferation in the draining lymph nodes at day 18 post-immunization (mean ± SEM; $n = 4$–5 mice per group; NS = non-significant, unpaired Student's $t$-test). **d** Representative histograms of CellTrace Violet proliferation for OT-II T cells taken from the draining lymph nodes between vehicle and MAZ51 treated EAE animals. **e** Quantitation of the average percent of OT-II T cell proliferation in the draining lymph nodes at day 18 post-immunization (mean ± SEM; $n = 4$–5 mice per group; $*p < 0.05$, unpaired Student's $t$-test)

CCR7$^+$ CD11c-eYFP$^+$ cell accumulation with CCL21 expression by lymphangiogenic vessels near the cribriform plate.

**Inhibition of VEGFR3 reduces CNS-derived antigen drainage**. We have demonstrated that the lymphangiogenic vessels near the cribriform plate can functionally carry CD11c-eYFP$^+$ cells, cells from the CNS parenchyma, and CSF (Fig. 4). Therefore, we asked whether these vessels can contribute to the drainage of CNS-derived antigens during EAE. We induced EAE in CNP-OP transgenic mice as previously described[52]; these mice express ovalbumin peptides under the CNP promoter[53] to drive ovalbumin peptide expression on oligodendrocytes. We then adoptively transferred ovalbumin specific, Thy1.1$^+$ congenic, CD8

(OT-I) and CD4 (OT-II) T cells labeled with CellTrace Violet. On day 18, the draining lymph nodes were harvested and the adoptively transferred cells were identified by their congenic Thy1.1 expression, and their proliferation measured by the dilution of CellTrace Violet (Fig. 6a, b). This transgenic system allows us to measure CNS-derived, ovalbumin-specific T cell proliferation as opposed to adoptively transferring MOG-specific 2D2 T cells, which may proliferate in response to MOG antigen drainage from both the CNS and immunization site. Beginning on day 7 post-immunization of EAE, mice were treated either with vehicle or 10 mg/kg of the VEGFR3 tyrosine kinase inhibitor MAZ51 once per day until sacrifice to determine if VEGFR3-dependent lymphangiogenesis contributes to CNS-derived oval-bumin antigen drainage (Supplementary Fig. 1). MAZ51 was

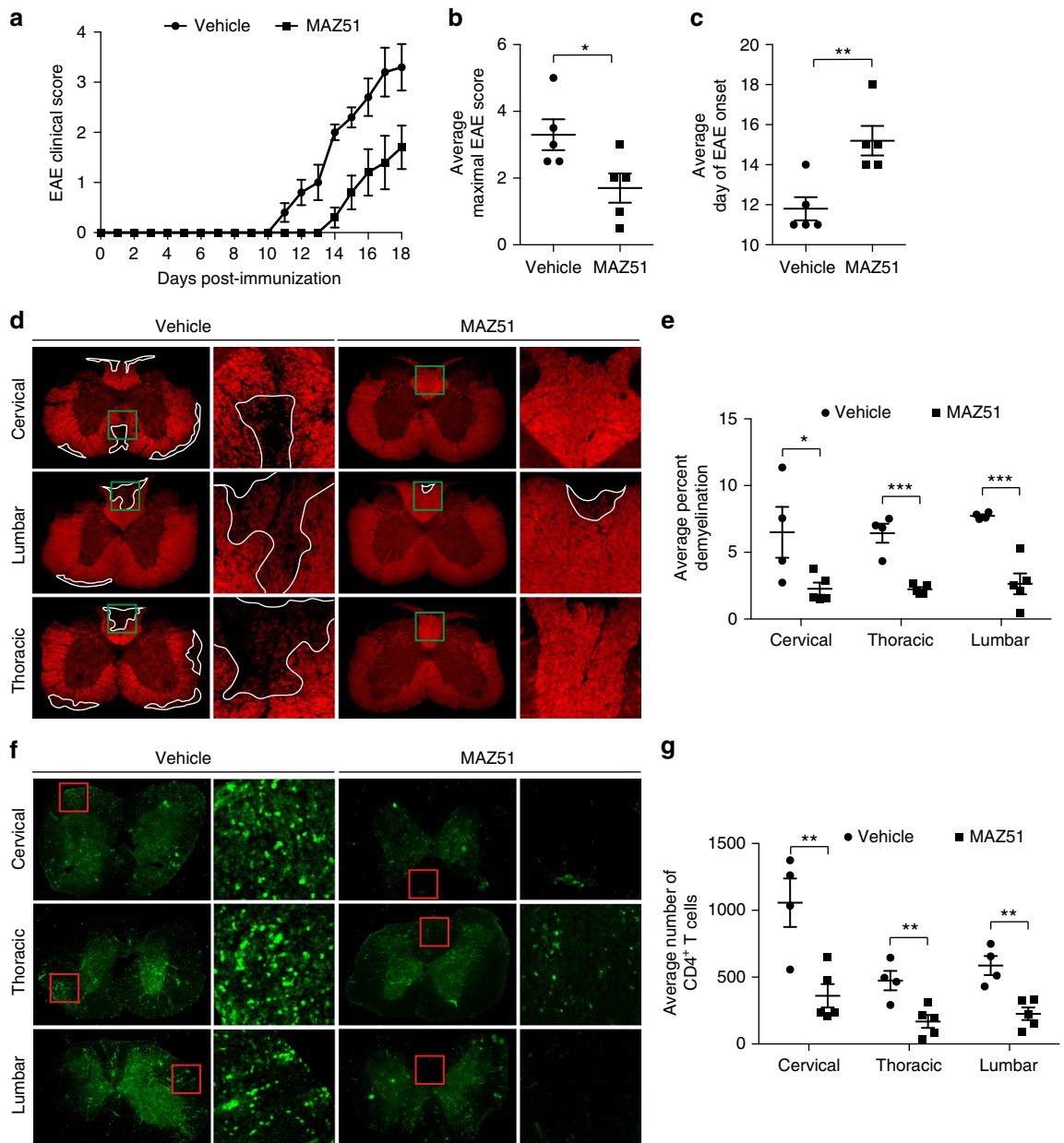

**Fig. 7** Inhibition of VEGFR3 reduces EAE severity. **a–c** ß-gal floxed ovalbumin peptides (OP) transgenic mice were crossed with an oligodendrocyte-specific Cre mouse (CNP-Cre) to generate CNP-OP transgenic mice. EAE was induced in CNP-OVA mice, treated with either vehicle or MAZ51, and were adoptively transferred CellTrace Violet labeled, Thy1.1 congenic, CD8 (OT-I), and CD4 (OT-II) T cells as shown in Fig. 5. EAE severity of mice receiving I.P. MAZ51 once per day beginning on day 7 post-immunization have reduced EAE severity (**a**), reduced average maximal EAE scores (**b**) ($n = 5$ mice per group; data are represented as mean ± SEM, *$p < 0.05$, unpaired Student's $t$-test), and delay in the EAE onset (**c**) ($n = 5$ mice per group; data are represented as mean ± SEM, **$p < 0.01$, unpaired Student's $t$-test). **d**, **e** The spinal cords of MAZ51 and vehicle treated animals were immunolabeled with Fluoromyelin to visualize myelin within the spinal cord (**d**). MAZ51-treated mice have significantly reduced percent demyelination when compared to vehicle-treated controls (**e**) ($n = 4$–5 mice per group; data are represented as mean ± SEM, *$p < 0.05$, ***$p < 0.001$, two-way ANOVA with Sidaks multiple comparisons test). **f–g** The spinal cords of MAZ51 and vehicle treated animals were immunolabeled with CD4 to visualize CD4+ T cell infiltration into the spinal cord (**f**). MAZ51-treated mice have significantly reduced CD4 T cell infiltration into the spinal cord when compared to vehicle-treated controls (**g**) ($n = 4$–5 mice per group; data are represented as mean ± SEM, **$p < 0.01$, two-way ANOVA with Sidaks multiple comparisons test)

administered beginning on day 7 post-immunization to prevent any unspecific effects in the initial priming and expansion of T cells, and has previously been demonstrated to not interfere with T cell proliferation in vitro[18]. It is also important to note that MAZ51 not only inhibits lymphangiogenesis but also causes meningeal lymphatic regression. However, it is technically difficult to inhibit specifically lymphangiogenic vessels near the

cribriform plate because of their anatomical location and the potential access to CNS-derived molecules by other lymphatics.

Quantitation of CellTrace Violet proliferation shows no significant changes in OT-I T cell proliferation (Fig. 6b, c), but there is a decrease in OT-II T cell proliferation in the draining lymph nodes of mice treated with MAZ51 during EAE (Fig. 6b–e). OT-I T cells have a much higher percentage of proliferation

(≈ 94% for vehicle, ≈ 85% for MAZ51) compared to OT-II T cells (≈26% for vehicle, ≈7% for MAZ51), which is likely attributed to the observation that the majority of OT-I T cells will proliferate even under steady-state conditions using this transgenic system[52]. This suggests that basal levels of CNS-derived antigen drainage is sufficient to drive OT-I T cell proliferation. Although non-significant, there is approximately a 10% reduction in OT-I T cell proliferation in the draining lymph nodes of MAZ51-treated mice compared to vehicle-treated controls, suggesting that VEGFR3 inhibition may partially correlate with reduced CD8 T cell proliferation (Fig. 6b, c). In contrast, OT-II T cells proliferate to a much lesser extent under both steady-state conditions and during EAE[52] (Fig. 6d–e), suggesting that increased CNS-derived antigen drainage is required to drive OT-II T cell proliferation. Furthermore, pharmacological inhibition of VEGFR3 is able to reduce CNS antigen-specific T cell proliferation in another transgenic system where ovalbumin peptides are expressed by neural progenitor cells instead of oligodendrocytes, confirming our results (Supplementary Fig. 8). These data suggest that VEGFR3 inhibition decreases the amount of antigen drainage to the draining lymph nodes, potentially due to inhibition of lymphangiogenesis. These data are also consistent with lymphangiogenesis contributing to increased fluid drainage from the CNS during EAE[54]. Meningeal lymphatics may also play a role in regulating CNS-derived antigen drainage as MAZ51 also induces meningeal lymphatic regression in both steady-state and EAE (Fig. 5). Nevertheless, the uniqueness of lymphangiogenesis to the cribriform plate during EAE implicates these vessels as a distinct site for the drainage of CNS-derived molecules. Therefore, both cribriform plate lymphatics and meningeal lymphatics may contribute to the drainage of CNS-derived antigens[32] and fluid.

**Inhibition of VEGFR3 reduces EAE severity**. Surprisingly, EAE mice treated with MAZ51 have lower disease severity and delayed EAE onset compared to vehicle-treated controls (Fig. 7a–c). MAZ51-treated animals also have reduced demyelination (Fig. 7d–e) and CD4 T cell infiltration (Fig. 7f–g) in the spinal cord. The reduction in EAE severity after treatment with MAZ51 is maintained when the EAE is monitored until day 25 post-immunization (Supplementary Fig. 9). Inhibition of VEGFR3 also reduces EAE severity in another transgenic system where oval-bumin peptides are expressed by neural progenitor cells (Sup-plementary Fig. 10), confirming our results. The reduction in EAE severity by VEGFR3 inhibition correlates with our prior observations of VEGFR3 contributing to lymphangiogenesis (Fig. 4), CNS antigen drainage (Fig. 6), and regression of meningeal lymphatics (Fig. 5), suggesting that VEGFR3 may regulate EAE severity through lymphangiogenesis and/or meningeal lymphatic regression. However, when MAZ51 is administered after the onset of EAE clinical symptoms (approximately 12 days post-immunization), the amelioration of EAE severity is abolished (Supplementary Fig. 9). These data identify VEGFR3 as a novel regulatory factor in the initiation of EAE, which coincides with its role in lymphangiogenesis and maintenance of meningeal lymphatics.

## Discussion
The drainage of CSF, antigens, and immune cells from the CNS may contribute to several neurological diseases including Alz-heimer's disease, lymphoedema, and multiple sclerosis[13,55–58]. Recently, meningeal lymphatics have been characterized during development and under steady-state conditions, and speculation about their role in neuroinflammation has generated considerable interest[12–14,59]. Lymphatic vessels near the cribriform plate have also been hypothesized to drain CSF, antigens, and cells from the

CNS[5–8,44,45,60]. However, there has been uncertainty as to whe-ther cribriform plate lymphatics reside on the CNS side of the cribriform plate or solely on the ventral side, and if they con-tribute to CNS immune diseases[8,27]. To our knowledge there have been no studies that have carefully defined cribriform plate lymphatics during neuroinflammation, characterized their ability to drain CSF and cells from the CNS parenchyma, established how they may regulate CNS autoimmunity, or investigated the role of VEGFR3 in the pathogenesis of EAE. Recently, it has been proposed that meningeal lymphatics regulate neuroinflamma-tion[32]. Here, we show for the first time that neuroinflammation induces VEGFR3-dependent lymphangiogenesis in the cribriform plate but not in meningeal lymphatics. These lymphangiogenic vessels in the cribriform plate can drain CSF and immune cells from the CNS parenchyma through a CCR7-dependent mechanism. We also demonstrate that lymphatic vessels in the nasal mucosa penetrate the cribriform plate and into the sub-arachnoid space. Our findings suggest that neuroinflammation-induced lymphangiogenesis near the cribriform plate contributes to fluid, cell, and antigen drainage. We also demonstrate that VEGFR3/VEGFC and the pre-existing lymphoid vessels near the cribriform plate contribute to these processes. The heterogeneity of CNS lymphatics in inflammation-induced plasticity warrants further studies to elucidate their contributions to neuroinflammation.

Several routes have been implicated in the drainage of fluid from the CNS, and the major route(s) of drainage from the CNS have been controversial[10,13,32,61,62]. Tracers injected into the cisterna magna suggest drainage occurs through lymphatics at the base of the brain, lymphatics near the cribriform plate, and along various cranial nerves[5–7,9,10,47,63–66]. Several of these reports attributed either minor or non-existent drainage into dural lymphatics[7,47]. Conversely, other studies have demonstrated that CSF infusion of various dyes and macromolecules can in fact drain into dural lymphatics[3,17,32], yet it is unknown how dural lymphatics access the CSF through the arachnoid barrier. Func-tional ablation of meningeal lymphatics reduces drainage to the cervical lymph nodes[17,32] and occlusion of the cribriform plate increases intracranial pressure[67], suggesting both routes may play a role in facilitating drainage from the CNS.

In this study we demonstrate that VEGFC is up-regulated within the inflamed CNS, and there is substantial accumulation of VEGFC producing macrophages and dendritic cells near the cribriform plate to induce lymphangiogenesis during EAE. These data suggest that lymphatics near the cribriform plate facilitate the drainage of either: (1) soluble VEGFC produced within the CNS, (2) the drainage of CNS-derived VEGFC-producing immune cells, (3) CNS-derived VEGFC to recruit systemic VEGFC-producing cells, and/or (4) CNS-derived antigens/anti-gen presenting cells to recruit systemic VEGFC-producing immune cells to the cribriform plate. Lymphangiogenesis occur-ring in vessels dorsal to the cribriform plate strongly implicate their role in the drainage of CNS-derived antigens, proteins, and/ or cells. It is also important to note that alternative pathways of drainage also exist such as meningeal lymphatics or along cranial nerves. Interestingly, lymphangiogenesis was not observed in meningeal lymphatics during EAE despite their implication in regulating EAE[32]. Lymphatic vessels on the CNS side of the cri-briform plate may have easier access to CNS-derived antigens, proteins, and/or cells due to the arachnoid barrier separating meningeal lymphatics from the CSF and the lack of an arachnoid barrier separating cribriform plate lymphatics from CSF[62,64].

The CNS parenchyma contain very few if any dendritic cells, which are primarily restricted to the meninges and perivascular spaces. However, during EAE there is a remarkable increase in monocytic dendritic cell migration into the CNS parenchyma[43].

It has previously been reported that the choroid plexus is the primary site of entry for dendritic cells during EAE, which are critical for re-stimulating antigen-specific effector T cells in situ during autoimmunity[68]. Additionally, they are able to carry antigens to the draining lymph nodes to prime antigen-specific T cells. Several groups have proposed that immune cells may migrate from the choroid plexus to the ventral regions of the olfactory bulbs and into cribriform plate lymphatics through the rostral migratory stream during EAE, although this has never been definitively shown[43–45]. Consistent with this hypothesis, we observed an accumulation of dendritic cells in the ventral regions of the olfactory bulbs during EAE, as well as near the cribriform plate and within cribriform plate lymphatics. These dendritic cells may also recruit other cells such as macrophages to increase VEGFC to contribute to lymphangiogenesis. This in turn may increase antigen access to the lymph nodes and promote fluid management. Therefore, we propose that neuroinflammation-induced lymphangiogenesis near the cribriform plate plays both an immunological role by increasing peripheral immune surveillance of the CNS as well as a physiological role by draining excess fluid. Nevertheless, how cells are able to traverse from the CNS parenchyma into lymphatics near the cribriform plate should be explored further.

Increased CNS antigen sampling in the draining lymph nodes have been shown in EAE models and MS patients[11]. Here, we report VEGFR3-dependent lymphangiogenesis near the cribriform plate during EAE and increased antigen drainage to the lymph nodes. These newly formed lymphatic vessels may promote CNS antigen-specific OT-II T cell priming, expansion, and EAE pathology. However, other T cells may require less antigen in which case baseline levels of lymphatics may be sufficient to induce priming and expansion as seen with OT-I T cells, which require 100-fold less antigen presentation than OT-II T cells for proliferation[69,70]. While this observation is restricted to the OT-I/OT-II system, these data suggest that it is possible that neuroinflammation-induced lymphangiogenesis can broaden the T cell repertoire by exposing lower affinity T cells to increased antigen presentation, impacting CNS autoimmunity.

Lymphangiogenesis has been characterized during peripheral infection and in tumors, in which two possible mechanisms have been proposed: (1) proliferation of pre-existing lymphatic endothelial cells[21,23,36–38] or (2) trans-differentiation of infiltrating monocytes[39–41]. In this study, we show an increased percentage of Ki67+ lymphatic endothelial cells near the cribriform plate during EAE, suggesting proliferation of pre-existing lymphatic endothelial cells may contribute at least partly to lymphangiogenesis. Whether or not trans-differentiating monocytes can contribute to lymphangiogenesis remains unclear. Interestingly, lymphatics near the cribriform plate contain higher levels of CD11c+ and/or CD11b+ cells during EAE. While our data implicate these cells as a source of VEGFC to promote VEGFR3-dependent lymphangiogenesis, we cannot exclude the possibility that some of these cells are able to trans-differentiate into lymphatic endothelial cells. Unfortunately, depletion studies of CD11b+ cells would fail to address this question as these cells are also needed to produce VEGFC to drive VEGFR3-dependent lymphangiogenesis. Future fate mapping studies are needed to elucidate the role of trans-differentiation.

In our studies, inhibition of lymphangiogenesis using MAZ51 resulted in a delay of EAE onset and reduced EAE severity. We utilized a transgenic system in which ovalbumin (OVA) peptides are expressed by oligodendrocytes, which causes pathogenic OVA-specific T cell proliferation in the draining lymph nodes to exacerbate EAE[52,71]. One possible explanation for the associated reduction in EAE severity in this model is that MAZ51 inhibits lymphangiogenesis at the cribriform plate and consequently

reduces the amount of CNS antigen drainage and pathogenic CNS antigen-specific T cell proliferation. However, the profound delay in EAE onset and reduction in severity suggest that other mechanisms may also play a contributing factor. It has previously been reported that sustained VEGFR3 activity is required for the maintenance of meningeal lymphatics, as pharmacological VEGFR3 depletion using sunitinib resulted in regression of these lymphatic vessels[17]. MAZ51 also causes meningeal lymphatic regression during EAE, suggesting that meningeal lymphatics may also play a role in the drainage of CNS-derived antigens[32]. Interestingly, MAZ51 has no effect on lymphatic vasculature near the cribriform plate during steady-state conditions. This is consistent with previous reports in which only dural lymphatics of the brain and to a limited extent those surrounding the spinal cord were susceptible to regression after VEGFR3 inhibition[17]. Further studies are needed to confirm the diffusion of MAZ51 and other VEGFR3 inhibitors in vivo in order to evaluate the therapeutic potential of lymphoid-modifying treatments in CNS autoimmune diseases. Additionally, it has also been hypothesized that the VEGFC–VEGFR3 axis may play a role in the recruitment of immune cells into the CNS[49], although this has never been definitively shown. Therefore, alternative mechanisms of VEGFR3 inhibition should also be considered. Nevertheless, our data identifies VEGFR3 as a novel factor in the initiation of EAE and as a potential therapeutic target.

It is unknown what are the relative route(s) of drainage in humans; however, it is hypothesized that arachnoid villi contribute to the drainage of CSF much more significantly in humans than in mice. Nevertheless, drainage of CSF through the cribriform plate have also been observed in humans[72–74]. Functionally, MRI and PET imaging have shown that Alzheimer's patients suffer from reduced drainage through the cribriform plate, presumably contributing to the accumulation of pathological proteins[75]. In addition to Alzheimer's disease, cribriform plate lymphatics and lymphangiogenesis may play a critical role in fluid homeostasis and the management of edema. Sealing the cribriform plate significantly reduces the rate of CSF absorption and increases intracranial pressure in sheep[67], suggesting that the extent of cribriform plate lymphatics may affect the drainage of extra fluid from the CNS during neuroinflammation. We speculate that brain immune responses can contribute to fluid management by regulating cribriform plate lymphatics, which may be relevant for several other neurological diseases. Our data suggest VEGFR3 may also play a novel role in the initiation of CNS autoimmunity, which may or may not depend on lymphangiogenesis.

In summary, we show for the first time that neuroinflammation induces VEGFR3-dependent lymphangiogenesis near the cribriform plate but not in meningeal lymphatics. Lymphangiogenesis induced by neuroinflammation near the cribriform plate functionally increases access of brain derived antigens to the draining lymph nodes as well as facilitates the drainage of excess fluid. Increased neuroinflammation-induced lymphangiogenesis can consequently affect CNS immune surveillance, T cell priming, and fluid management to influence the outcome of neuroinflammatory diseases. A better understanding of brain drainage during neuroinflammation can lead to novel therapeutic approaches.

## Methods
**Animals**. Male and female C57BL/6J wild-type, C57BL/6-Tg(Prox1-tdTomato) 12Nrud/J, B6.PL-Thy1a/CyJ (Thy1.1 congenic mice), C57BL/6-Tg(TcraTcrb) 1100Mjb/J (ovalbumin specific CD8 T cell or OT-I mice), B6.Cg-Tg(TcraTcrb) 425Cbn/J (ovalbumin specific CD4 T cell or OT-II mice), B6.Cg-Tg(Nes-Cre)1Kln/J, and Tg(CAG-KikGR)33Hadj/J mice were all purchased from Jackson Laboratories. B6.Cg-Tg(Itgax-Venus)1Mnz/J (CD11c-eYFP mice) were a generous gift from Michel C. Nussenzweig at Rockefeller University. Cnp-Cre transgenic mice

were a generous gift from Brian Popco at the University of Chicago. pZ/EG-OP OVA$_{257-264}$-OVA$_{323-339}$ (Ovalbumin floxed mice) were generated by our lab as previously described in the C57BL/6J background[49] (Supplementary Table 1).

pZ/EG-OP OVA$_{257-264}$-OVA$_{323-339}$ (ovalbumin floxed mice) were crossed with CNP$^-$Cre mice to generate CNP-Ovalbumin Peptide (CNP-OP) mice that express OVA$_{257-264}$ under the CNS oligodendrocyte-specific CNP promoter[53] or crossed with Nes-Cre mice to generate Nestin-Ovalbumin Peptide (Nes-OP) mice that express OVA$_{257-264}$ and OVA$_{323-339}$ under the CNS neural progenitor-specific Nestin promoter. Both C57BL/6-Tg(TcraTcrb)1100Mjb/J (Ovalbumin-specific CD8 T cell or OT-I mice) and Tg(TcraTcrb)425Cbn/J (Ovalbumin-specific CD4 T cell or OT-II mice) were each crossed with B6.PL-Thy1a/CyJ (Thy1.1 congenic mice) to generate Thy1.1 congenic ovalbumin-specific CD4 and Thy1.1 congenic ovalbumin-specific CD8 T cell mice. These congenic ovalbumin-specific T cells were then adoptively transferred into the CNP-OP mice or Nes-OP mice for proliferation experiments.

Eight to twelve-week-old female mice were used for all EAE experiments along with the appropriate age- and sex-matched littermate controls. All experiments were conducted in accordance with guidelines from the National Institutes of Health and the University of Wisconsin-Madison Institutional Animal Care and Use Committee.

**EAE induction.** EAE was induced in 8–12-week-old female mice by subcutaneous immunization with 100 μg of MOG$_{35-55}$ emulsified with Complete Fruend's Adjuvant (CFA)[76] (Supplementary Table 1) between the shoulder blades. Two hundred nanogramsof Pertussis Toxin (PTX) was injected intraperitoneally at 0 d. p.i. and 2 d.p.i.[70] (Supplementary Table 1). The onset of clinical scores were observed between day 8 and 12 post-immunization, and were assessed daily as follows: 0, no clinical symptoms; 1, limp/flaccid tail; 2, partial hind limb paralysis; 3, complete hind limb paralysis; 4, quadriplegia; 5, moribund. Intermediate scores were also given for the appropriate symptoms. For a detailed list of EAE reagents, refer to Supplementary Table 1.

**Photoconversion of KikGR transgenic mice.** For intracerebral photoconversion, KikGR transgenic mice were anesthetized with isoflurane and the head shaved. A Zibra Corp custom 405 nm light source (Product # 50146) coupled to a custom fiber optic quartz light bundle (Product # 50145) was funneled through an 18-gauge short bevel needle with a penetrating depth of 1.55 mm from the surface of the brain. The intracerebral photoconversion was restricted to the ventral–posterior region of the frontal lobe. Photoconversion consisted of 5-min exposures, once per day at 15 d.p.i and 16 d.p.i. of EAE. Mice were harvested at 17 d.p.i. to allow visualization of photoconverted cells after 24 and 48 h (Supplementary Fig. 1). The heads were fixed in 4% PFA overnight, which although lowers the photoconverted signal intensity of the Kikume protein, is still obvious by microscopy[77,78]. Additionally, fixation is required prior to decalcification. This protocol causes a localized photoconversion within the CNS parenchyma, yielding approximately 7% photoconversion of cells by area per brain section.

**Evans blue dye injections.** For experiments showing Evans blue dye can drain from the CSF into cribriform plate lymphatics, 10 μl of 10% Evans blue dye was injected into the cisterna magna at a rate of 2 μl/min. After allowing 30 min for the dye to circulate, the mice were euthanized and the whole heads analyzed for Evans blue dye distribution by microscopy (Fig. 3).

**Magnetic resonance imaging.** MRI experiments were done using a 4.7 T small animal MRI (Agilent Technologies Inc., Santa Clara, CA, USA) and acquired using VnmrJ (Agilent Technologies). After scout scans, isotropic 3D T1-weighted scans were used to detect gadolinium using the following parameters: TR = 9.3 ms; TE = 4.7 ms; flip angle = 20°; field of view = 40 × 20 × 20 mm; resolution = 256 × 128 × 128; averages = 4; voxel size≈156 μm³. These resulted in a time scan of approximately 11 min. Animals were anesthetized using isoflurane administered through a nose-cone, and 10 μl of gadolinium was injected into the cisterna magna at a rate of 2 μl/min. Respiratory rates were monitored to ensure normal physiology. A baseline scan was acquired prior to gadolinium injection, and image processing was done using FIJI software.

**Lymphocyte isolation.** Mice were anesthetized with ketamine/xylazine and transcardially perfused with phosphate-buffered saline (PBS). Single-cell suspensions were made from spleens, cervical lymph nodes, and brains after digestion by pushing cells through a 70 μm nylon cell strainer with a 1 ml syringe plunger. Red blood cells were lysed using ACK lysis buffer (0.15 M NH$_4$Cl, 10 mM KHCO$_3$, 0.1 mM EDTA), and washed with HBSS + 10% FBS. Brain cell suspensions were then digested with 1 mg/ml of Collagenase D and 28 U/ml of DNAse I in a 37 °C incubated shaker for 45 min. Brain cell suspensions were then resuspended in 30% Percoll and layered above 50% Percoll to generate a Percoll gradient. The gradient was centrifuged at 1258 rcf for 30 min without brake. The interface was removed and washed with HBSS. All organs were weighed and live cells were counted using a hemocytometer.

**Flow cytometry.** Single-cell suspensions from all organs were resuspended and washed in fluorescence-activated cell sorting (FACS) buffer (pH 7.4, 0.1 M PBS, 1 mM EDTA, 1% BSA), then immunolabeled with the appropriate conjugated antibodies (Supplementary Table 1) for 30 min at 4 °C. Stained cells were then washed with FACS buffer and fixed overnight in 4% PFA (4% paraformaldehyde, 0.1 M PBS). Primary antibodies used for flow cytometry were used as follows: rat anti-CD4-FITC-conjugated (1:200; 553047; BD Pharmingen), rat anti-CD4-AF647-conjugated (1:200; 557681; BD Pharmingen), rat anti-CD8-FITC-conjugated (1:200; 553031; BD Pharmingen), rat anti-CD8-APC-conjugated (1:200; 553035; BD Biosciences), rat anti-Vß5.1, 5.2-PE-conjugated (1:200; 553190; BD Pharmingen), rat anti-CD90.1-PE-conjugated (1:200; 202523; Biolegend), and rat anti-CD90.1-APC-Cy7-conjugated (1:200; 561401; BD Biosciences). Data were acquired using a BD LSR II Flow Cytometer (BD Biosciences) and analyzed using FlowJo software. For a detailed list of reagents and software refer to Supplementary Table 1.

**Histology.** Mice were anesthetized with ketamine/xylazine and transcardially perfused with PBS followed by perfusion with 4% PFA. Mice were then decapitated and the skin was removed from the whole heads using forceps and scissors to separate the skin from the muscle and ear canal. The whole heads were fixed in 4% PFA overnight. The whole heads were then decalcified in 14% EDTA for 7 days followed by cryoprotection in 30% sucrose for 3 days (Supplementary Fig. 1). The EDTA was replaced with fresh 14% EDTA each day. For whole spinal column preparations, the skin was removed and the spinal column separated from the rib cage. The spinal columns were then fixed in 4% PFA overnight, decalcified in 14% EDTA for 7 days followed by cryoprotection in 30% sucrose for 3 days. The decalcified mouse heads and spinal columns were then embedded in Tissue-Tek OCT Compound, frozen in dry ice, then stored at −80 °C. Sixty-micrometer-thick frozen sections were obtained on a Leica CM1800 cryostat, mounted on Superfrost Plus microscope slides, and stored at −80 °C.

**Immunohistochemistry and confocal microscopy.** For immunohistochemistry, sections were thawed at room temperature for 10 min, washed with PBS for 10 min, then blocked in 10% BSA for 60 min. Sections were then incubated with the appropriate primary antibodies in 0.1% Triton-X in 0.1 M PBS at 4 °C overnight in a humidified chamber. Sections were then washed three times with PBS for 10 min each, then incubated with the appropriate secondary antibodies at room temperature for 120 min in a humidified chamber. Sections were then washed three times for 10 min each, mounted with Prolong Gold mounting medium with DAPI, and images acquired using an Olympus Fluoview FV1200 confocal microscope. The brightness/contrast of each image was applied equally across the entire image and equally across all images and analyzed using FIJI software. Primary antibodies used for immunohistochemistry were used as follows: rat anti-Lyve-1-eFluor-570-conjugated (1:100; 41-0443-82; eBioscience), rat anti-Lyve-1-eFluor-660 conjugated (1:500; 50-0443-82; eBioscience), rat anti-Podoplanin-eFluo488 conjugated (1:100; 53-5381-80; eBioscience), goat anti-VEGFR3-unconjugated (1:100; AF743-SP; R&D Systems), rat anti-CD11b-PE conjugated (1:100; 553311; BD Pharmingen), rabbit anti-VEGFC-unconjugated (1:100; ab83905; Abcam), goat anti-CCL21-unconjugated (AF457-SP; R&D Systems), rat anti-CCR7-PE conjugated (560682; BD Pharmingen), rat anti-CD4-FITC-conjugated (1:100; 553047; BD Pharmingen), and Fluoromyelin Red Fluorescent Myelin Stain (1:1000; F34652; Thermo Fisher Scientific). The appropriate secondary antibodies were used as follows: donkey anti-goat-AF488-conjugated (1:500; A11055; Thermo Fisher Scientific), donkey anti-goat-AF405 (1:500; ab175664; Abcam), and donkey anti-rabbit-AF405 (1:500; ab175649; Abcam).

For a detailed list of reagents refer to Supplementary Table 1.

**Western blotting.** Mice were anesthetized with ketamine/xylazine and transcardially perfused with PBS. The brain and spinal cords were quickly harvested. The brain and spinal cords were dounced using a tissue douncer in RIPA buffer (pH 7.5, 25 mM Tris-Cl, 150 mM NaCl, 1 mM EDTA, 1% Triton X-100, 0.1% SDS) and the appropriate dilution of protease and phosphatase inhibitors. The dounced tissue was then sonicated and stored at −80 °C. Protein levels were normalized using a Bradford Assay and were assessed with Li-Cor Odyssey CLx infrared imaging system. Primary antibodies used for western blot were used as follows: goat anti-CCL21-unconjugated (1:100; AF457-SP; R&D Systems), rabbit anti-CCR7-unconjugated (1:100; ab32527; Abcam), rabbit anti-VEGFC-unconjugated (1:100; ab83905; Abcam), and chicken anti-ß-actin-unconjugated (1:1000; ab13822; Abcam). The appropriate secondary antibodies were used as follows: donkey anti-rabbit IgG-IRDye 800CW (1:2000; 926–32213; Li-Cor), donkey anti-goat IgG-IRDye 800CW (1:2000; 925–32214; Li-Cor), and donkey anti-chicken IgG-IRDye 680LT (1:2000; 926-68028; Li-Cor). For a detailed list of reagents, refer to Supplementary Table 1. Representative uncropped western blots for VEGFC and ß-actin can be found in Supplementary Fig. 11.

**Adoptive transfers and drug administration.** Congenic Thy1.1, ovalbumin-specific OT-I and OT-II cells were isolated from the spleen, labeled with CellTrace Violet per manufacturer's instructions, and intravenously injected into host mice (2.0 × 10⁶ of each cell type). MAZ51 was dissolved in DMSO and intraperitoneally

injected at 10 mg per kg of mouse weight. Control mice received intraperitoneal injections of equivalent volumes of DMSO. MAZ51 or vehicle was administered beginning either on day 7 post-immunization of EAE before the onset of EAE clinical scores or after the first sign of EAE clinical score (approximately day 12 post-immunization.)

**Statistical analysis.** Average Lyve-1$^+$ vessel volume, western blots, percent T cell proliferation, cell numbers, day of EAE onset, and maximal EAE clinical scores were compared between two groups using unpaired Student's $t$-test. Two-way ANOVA using the Sidaks multiple comparison test was used to compare the number of CD4 T cells and percent demyelination in the cervical, thoracic, and lumbar spinal cord between vehicle and MAZ51 treated mice. One-way ANOVA using Tukeys multiple comparisotest was used to compare the average Lyve-1$^+$ vessel area between healthy and EAE vehicle versus MAZ51 treated mice for both the dura and cribriform plate. Statistical analysis was performed with GraphPad Prism 6.0 Software. For all statistical tests the significance is portrayed as: NS = non-significant, $p > 0.05$; $^*p < 0.05$; $^{**}p < 0.01$; $^{***}p < 0.001$.

**Reporting summary.** Further information on experimental design is available in the Nature Research Reporting Summary linked to this article.

## Data availability

A reporting summary for this Article is available as a Supplementary Information file. All data supporting the findings of this study are available from the corresponding authors upon reasonable request.

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

## Acknowledgements

We thank Khen Macvilay for his assistance in flow cytometry, Laura Schmitt-Brunold for her assistance in PCRs, and all members of our laboratory for helpful discussions and constructive criticisms of this work. Additionally, we would like to thank Dorian McGavern and Dritan Agalliu for insightful discussions and constructive criticisms. We would also like to thank the University of Wisconsin Translational Research Initiatives in Pathology laboratory, in part supported by the UW Department of Pathology and Laboratory Medicine and UWCCC grant P30 CA014520 for use of its facilities and services. We also thank Beth Rauch and Elizabeth Meyerand at the UW Small Animal Imaging Facility (SAIF) supported by the UWCCC grant P30 CA014520 for use of its facilities and services. This work was supported by National Institutes of Health grants NS108497 and NS103506 awarded to Z.F., HL128778 awarded to M.S., AI101378 awarded to W.J.K., and the Neuroscience Training Program T32-GM007507 (to M.H. and A.R.), and the American Heart Association award 15PRE25500022 (to A.R.).

## Author contributions

M.H. performed all of the experiments, analyzed the data, contributed to experimental design, generated the figures including schematics such as those shown in Fig. 1a, and wrote the manuscript. A.R. and J.A.K. bred the CNP-OP, OT-I Thy1.1, OT-II Thy1.1, and Nes-OP transgenic mice and assisted with the experiments and analysis of the data for Fig. 6 and Supplementary Fig. 8. Y.H.C. assisted with the experiments for Supplementary Fig. 6. J.H. titrated and optimized MAZ51. S.A.M. bred the KikGR transgenic mice and optimized the photoconversion for in vivo use. M.S., Z.F., and W.J.K. contributed to the experimental design and assisted in writing the manuscript. All authors edited the manuscript.

## Additional information

**Competing interests:** The authors declare no competing interests.

