## [Peer Review File · Nature Communications]

Reviewers' comments:

Reviewer #1 (Remarks to the Author):

In their manuscript "In situ neo-lymphangiogenesis induced by neuroinflammation near the cribriform plate contributes to the drainage of CNS derived antigens, immune cells, and CSF," Hsu et al. describe an incredibly interesting finding, that lymphatic endothelial cells proliferate during neuroinflammation and may play a role in antigen presentation and drainage of immune cells. Additionally, they show that this neo-lymphangiogenesis is VEGFR3/VEGFC-dependent. These data answer an important question in the field. Although many of the experiments were well-done, there are several major issues that need to be addressed before publication. In particular, it is important that the authors provide some sort of 3D view of the proliferating lymphatic endothelial cells to show that these cells actually form new vessels, rather than hypertrophy of existing vessels or proliferation of single cells. Additionally, the authors should conduct parallel analyses of other lymphatic vessels (such as the dura in the SC) with the same vigor as those in the cribriform plate, particularly in the VEGFR3 inhibition experiments.

Major issues

1. In Figure 1, it is clear that there are several Prox1+Lyve+ single cells aside from the tube shown in the center of the images. These should at least be addressed and explained in the text.
2. It is not clear that the extra Lyve1+ cells are making extra vessels, enlarging current vessels, or whether it is just a proliferation of the many single cells seen in Figure 1. This is a key component of the paper's main finding and should not be overlooked. The authors need to include either a 3D-reconstruction (eg. thick sections imaged with a confocal microscope) to more thoroughly describe the structure formed by the new cells.
3. The analysis of lymphatic vessels in the dura and spinal cord in Supplementary Figure 3 should be moved to the main paper, and analysis of these lymphatic vessels should be as thoroughly performed as for the cribriform plate lymphatic vessels. It is surprising and interesting that the inflammation-induced neo-lymphangiogenesis is specific to the cribriform plate.
4. In Figure 2, it is not clear that the Ki67 is co-localizing with the lymphatic vessels. Figure 2T looks very similar to Figure 3S; it might be that the Ki67+ cells are proliferating immune cells in the vessels rather than the vessels themselves. Performing FACS analysis on the lymphatic endothelial cells using the Prox-1-tdTomato mouse would be a better way to quantify markers of proliferation.
5. The authors' claims regarding CCL21/CCR7 should either be entirely removed, proven, or their language should be changed in every instance describing this data. The data presented in Figure 5 do not lead to the conclusion stated in the title of the figure.
6. MAZ51 needs to be administered to healthy mice to determine whether VEGFR3 is required for maintenance—if the drug is getting rid of existing vessels, this would affect the presented analysis.
7. The effect of MAZ51 on other lymphatic vessels (such as the dura presented in Supplementary Figure 3) needs to be observed and quantified. The phenotype in the EAE experiments using this drug may be due to an effect on all lymphatics.

Minor issues

1. Figure 2 A and B are too dim and do not show the extent of proliferation of lymphatic endothelial cells as convincingly as in Figure 3.
2. The authors state that they cannot determine the identity of the cells within the lymphatic vessels. They need to further discuss why this is the case and what negative results they have gotten. Their conclusion that they are not neurons or other CNS-resident cells is not necessarily valid—for example, the cells might be neurons killed by the photoconversion. Furthermore, their conclusion that the new vessels are functional does not follow from the data presented—there is

no evidence that the vessels transporting cells and CSF are the new lymphatic vessels rather than the existing ones.

3. In their discussion of the data in Figure 4, the authors should mention the many VEGFC+ cells that are not CD11b+.
4. The authors need to quantify and discuss the clear increase in non-proliferating cells in MAZ51-treated OT-I T-cells in Figure 6.
5. It would be very interesting to see an extension of the EAE clinical score curve presented in Figure 7. While it is evident that the treatment delays onset of disease, it would be interesting to see whether the MAZ51-treated mice eventually catch up to the control mice in disease severity or whether the curve peaks at a lower value.

Typos/Mistakes

1. On page 5, "Vascular Endothelial Growth Factor 3" is missing the word "Receptor"
2. In Figure 2 C, D, F, and G, the image labels are mixed up.
3. In Figure 3, the images say "C11" instead of "CD11"

Reviewer #2 (Remarks to the Author):

The manuscript # NCOMMS-18-16010 addresses an important question in the field of neuroimmunology regarding the mechanisms of immune cell trafficking into and outside the CNS in neuroinflammation. The authors have previously shown that dendritic cells use the CCL21/CCR7-mediated signaling pathway to traffic outside the CNS in the EAE model of human MS. In this study, they focus on a group of lymphatic vessels surrounding the olfactory mucosa and cribriform plate and demonstrate that these lymphatics become hyperproliferative at EAE score 3, respond to VEGFC which is secreted in the nose region. Moreover, blockade of VEGFC/VEGFR3 signaling using a pharmacological inhibitor MAZ-51 reduced lymphatics, disease severity after the adoptive EAE transfer and results in decreased immune cell infiltration (dendritic cells and T cells) into the CNS.

This study is highly significant since it is unclear how the lymphatics surrounding the CNS can contribute to disease pathogenesis. Although VEGFC/VEGFR3 signaling has been shown to contribute to the development and maintenance of the CNS lymphatics, their role in CNS disease pathogenesis and neuroinflammation is unclear. The study is very rigorous and the quality of the data is very high. Nevertheless, there are a few concerns that the authors need to address to consolidate some of their findings:

- 1) The authors show by immunofluorescence and Western blotting that both VEGFC and CCL21 are upregulated in the olfactory mucosa in close proximity to the position of the lymphatics (Figs. 4 and 5). However, it is unclear which cells are secreting these factors based on the images since these are secreted factors. The authors should resolve this issue by performing in situ hybridizations with probes for VEGFC and CCL21.
- 2) When does VEGFC get induced during disease progression to promote lymphatic proliferation? The time course of VEGFC upregulation (mRNA and protein) needs to be better described in relation to disease progression (Score 1-3) using both qRT-PCR and western blotting.
- 3) Does administration of MAZ51 influence the lymphatics in other regions of the CNS? Although these lymphatics may not change in disease progression (EAE) as the authors have nicely shown in the supplementary data, they may be affected by MAZ51 and influence immune cell trafficking into the CNS.

4) The kinetics of Evans blue flow through the cribriform plate into the nasal cavity and deep cervical lymph nodes need to be better determined in control mice, EAE score 3.0 and EAE score 3.0 mice with MAZ51. The authors should measure dynamically over time the flow of Evans blue dye to provide convincing evidence that these lymphatics increase the drainage.

5) The authors show in Figure 3 that some photoconverted cells exit the CNS and some remain in the parenchyma. What is the fraction of photoconverted cells that exit the CNS versus those that remain in the parenchyma? This is an important information for the study.

Reviewer #3 (Remarks to the Author):

In the manuscript of Hsu et al, the authors have provided a characterization of cribriform plate (CP) lymphatic vessels and have demonstrated that they can expand during EAE. The authors have shown evidence of monocyte-derived cells producing VEGF-C in close proximity to LVs. An inhibitor of VEGFR3 led to a reduced expansion of LVs and a delayed time course of EAE. This is an interesting report that suggests that the more traditional view of CSF lymphatic outflow through the cribriform plate may be more important, at least for the initiation of EAE, than the recently rediscovered dural lymphatic vessels. However, there are many places where additional data may be beneficial to strengthen the conclusions of the study.

Major comments:

1. The authors conclude that dendritic cells are migrating through the CP rather than through dural LVs, but no evidence is presented showing a lack of cells within the dural LVs. In addition, the data (Fig 3O-S) that is shown purporting that cells are within CP lymphatics is dubious, as there are no orthogonal views showing that cells are truly within the vessels rather than sitting upon them. Indeed, there are many cells that are distributed in this region that are not associated with LVs. Wouldn't it be possible that cells could also exit through extravascular pathways through the plate to reach lymphatic vessels within the nasal cavity? A recent study has shown after intranasal administration, stem cells can reach the SAS through routes that are not associated with lymphatic vessels or the olfactory nerves (Galeano et al, Cell Transplantation, 2018, DOI: 10.1177/0963689718754561).
2. There is a delayed onset of EAE but there appears to be no difference in progression once the disease has initiated. Could the authors speculate on why this is the case? It would be important to assess whether treatment with MAZ51 induced a regression of either dural LVs or CP LVs. As mentioned by the authors in the discussion, Antila et al, JEM, 2017 suggest that regression of dural LVs would take place with treatment of a tyrosine kinase inhibitor (Sunitinib) or soluble VEGFR3. One would expect that MAZ51 would also induce a regression of these vessels. It is important to try to elucidate whether the delay in EAE due to MAZ51 treatment is due to a lack of expansion of CP LVs or a regression of dural LVs (or a possible combination of both).
3. The functional data presented is not strong. 10 uL of 10% Evans Blue is not an appropriate tracer administration to assess CSF outflow. For one, murine CSF volume is estimated to be around 35 uL, so 10 uL is too high of a volume to administer. In addition, Evans Blue at such a high concentration likely has toxic effects in the CNS and an uncertain percentage will also bind to tissues and proteins within the SAS. A fluorescently labeled dextran (or better yet albumin) would be a more suitable tracer to use. The images presented in Supp Fig 5 do not support the claims of the authors that flow has occurred "along the superior sagittal sinus, surrounding the olfactory bulbs and flowing into the nasal cavity" One would expect that the bulk of the flow towards the CP would occur in the SAS on the ventral side of the brain rather than at the dorsal side near the SSS (for visualization see Pizzo et al, Physiology, 2018, DOI: 10.1113/JP275105). Also, the image shown in Supp Fig 5 of the LNs appears to show only superficial cervical LNs, not deep cervical LNs. According to Ma et al, Nat Comm, 2017 (DOI: 10.1038/s41467-017-01484-6), the lymphatic route to the superficial cervical LNs (termed mandibular LNs by these authors) would occur due to outflow along optic nerves. The large amount of EB around the eyes apparent in the image in Supp

Fig 5 would be supportive of this outflow pathway also playing a role.

4. In order to have a clinically meaningful result the authors should perform an intervention study to determine how the progression of the disease is affected after it has already been initiated. A prevention study as is shown in the paper would not be possible in patients with MS. The authors mention several times that lymphangiogenesis may manage fluid accumulation during neuroinflammation. In this case, an inhibitor of lymphangiogenesis would not be beneficial. One would expect perhaps that the disease would worsen if fluid and inflammatory factors were allowed to accumulate within the CNS, similar to what occurs in models of psoriasis, rheumatoid arthritis or chronic airway inflammation that are cited by the authors.

5. The authors need to provide more compelling evidence showing that cells have truly migrated from the parenchyma to the CP lymphatics. How do the authors envision that this takes place? If migration truly occurs through the rostral migratory stream as suggested by the authors then fluorescently switched cells should be found along this stream leading to the olfactory bulb. How would the cells then transverse the SAS to reach the lymphatics?

6. The authors attempt to make the claim, without any supporting data that monocyte derived cells may be incorporating into lymphatic vessels through trans-differentiation. This is a controversial topic. Since the original reports in 2005 and 2006, several groups have attempted to lineage trace cells that are contributing to the expansion of lymphatic vessels. The conclusion that has been reached is that monocytes are likely not a major contributor to this process. If the authors want to make this claim they will need to provide evidence supporting this.

Minor comments:

1. Please use scatter plots rather than bar graphs to show results.

2. Please avoid use of the prefix "neo" within the term "neo-lymphangiogenesis".

Lymphangiogenesis refers to the expansion of an existing lymphatic network, which is what is suggested in the study.

3. LVs were identified "to penetrate from the middle nasal turbinate through the cribriform plate and into the subarachnoid space" How is it shown that these vessels are in direct contact with the SAS? If perineural sheaths extend along olfactory nerves as has been described in the past, then why would there be a need for a gap in the arachnoid barrier above the plate? On the other hand, if no barrier exists at this location than cells should be able to cross the plate with ease and would not need lymphatic vessels to do so.

4. Supp Fig 3 shows dural LVs close to olfactory bulbs with no expansion during the progression of EAE. Wouldn't these vessels be expected to be connected to the LVs at the CP?

5. Supp Fig 4 and 5 are presented as evidence that there is no lymphangiogenesis occurring in the dural or spinal LVs, however, no quantifications are made of these vessels to support this.

6. The authors write in several places throughout the text that historically it is well-established that the majority of CSF flows through the cribriform plate. This is not, at the present time, the accepted dogma in the field. The dogma has persisted that arachnoid villi or projections are the major outflow route with lymphatics playing a supporting role. Since the high-profile publications describing the meningeal lymphatic vessels, this lymphatic network has also been considered to take a major part in CSF outflow. Previous studies have quantified outflow within cervical LVs and have recovered around 30 to 50% of tracers within these vessels. No study made the claim that CSF outflow predominantly occurs through lymphatic vessels until last year (Ma et al, Nat Comm, 2017). Of note, this study found several routes for outflow in addition to the nasal route.

7. References 16 and 17 are not the correct ones for the statement in the Introduction that LECs proliferate and undergo VEGFR3 dependent lymphangiogenesis in the meninges.

8. Introduction: "Several neuroinflammatory diseases, including MS and EAE" – these are not different diseases, one is a model for the other.

9. Results: "trauma induced by intracerebral injection" – what injection are the authors referring to?

10. Discussion: "Previous murine studies have also demonstrated that the majority of CSF

drainage occurs through the cribriform plate" – Refs given are from the Johnston group who worked primarily with rat and sheep.

11. Discussion: Several of the references given for CSF drainage through the CP in humans are not supportive of this concept. For example, reference 64 is a report on LVs in the intravertebral disc. Reference 66 discusses drainage from the nasal tissue not the CNS.

12. Labels on Fig 2 should read "CD11c" rather than "C11C"

Reviewer #1 (Remarks to the Author):

In their manuscript “In situ neo-lymphangiogenesis induced by neuroinflammation near the cribriform plate contributes to the drainage of CNS derived antigens, immune cells, and CSF,” Hsu et al. describe an incredibly interesting finding, that lymphatic endothelial cells proliferate during neuroinflammation and may play a role in antigen presentation and drainage

of immune cells. Additionally, they show that this neo-lymphangiogenesis is VEGFR3/VEGFC-dependent. These data answer an important question in the field. Although many of the experiments were well-done, there are several major issues that need to be addressed before publication. In particular, it is important that the authors provide some sort of 3D view of the proliferating lymphatic endothelial cells to show that these cells actually form new vessels, rather than hypertrophy of existing vessels or proliferation of single cells. Additionally, the authors should conduct parallel analyses of other lymphatic vessels (such as the dura in the SC) with the same vigor as those in the cribriform plate, particularly in the VEGFR3 inhibition experiments.

Reviewer 1, Major Concern #1: In Figure 1, it is clear that there are several Prox1+ Lyve1+ single cells aside from the tube shown in the center of the images. These should at least be addressed and explained in the text.

We thank the reviewer for pointing this out. Higher magnification of these images revealed these signals as non-cellular, non-specific, and localized to the outer layers of the olfactory bulbs. The outer layers of the olfactory bulbs, particularly the outer nerve layer and glomerular layer are sites of constant neuronal axon/synapse turnover from olfactory sensory neurons residing in the olfactory mucosa, and we suspect that this may lead to some unspecific autofluorescence. Other groups have independently reported autofluorescence in this region as well (**See reference #29: Chartoff, E.H., et al.,**

Stem Cells Int, 2011. 2011: p. 586586). We have included this reference as reference #29 and clarified this in the results section (page 5, lines 119 - 120):

“We also noticed non-cellular unspecific labeling of Prox1/Lyve-1 near the outer layers of the olfactory bulbs, potentially due to autofluorescence in this region²⁹.”

Reviewer 1, Major Concern #2: It is not clear that the extra Lyve-1+ cells are making extra vessels, enlarging current vessels, or whether it is just a proliferation of the many single cells seen in Figure 1. This is a key component of the paper’s main finding and should not be overlooked. The authors need to include either a 3D-reconstruction (e.g. thick sections imaged with a confocal microscope) to more thoroughly describe the structure formed by the new cells.

We strongly agree with the reviewer that lymphangiogenesis is the focus of this paper and clarification of the precise anatomy of the lymphangiogenic vessels has been addressed. In order to highlight the lymphangiogenic vessel anatomy, we have modified Figure 2 by separating and brightening the Lyve-1 channel from the CD11c-eYFP and DAPI channels in Figure 2A and Figure 2B for easier visualization (**Revised Figure 2A and 2B are shown below, and are part of a larger Revised Figure 2**). Additionally, each one of these images represent a z-stack of a relatively thick (60 micron) section; we have also included in the supplementary figure a 3D reconstruction after acquiring the images on a confocal microscope of section #6 in Figure 2A and section #5 in Figure 2B, which represent the sections with the largest Lyve-1⁺ vessel area in each group (see new 3D reconstruction videos in Supplementary Figures 7 and 8).

In response to the reviewer, we have completed the requested 3D reconstruction of lymphatic vessels near the cribriform plate between healthy and EAE (see new 3D reconstruction videos in Supplementary Figures 7 and 8), and clarified the observed changes in the results section of the text (page 6, lines 142-145):

“More specifically, we observed both an expansion of pre-existing lymphatic vessels and growth of lymphatic vessels both dorsally and laterally. Lymphangiogenesis seems to be localized to the CNS side of the cribriform plate. These observations can alternatively be visualized through higher magnification or 3D reconstruction (Supplementary Figure 3, 4, 5).”

Reviewer 1, Major Concern #3: The analysis of lymphatic vessels in the dura and spinal cord in Supplementary Figure 3 should be moved to the main paper, and analysis of these lymphatic vessels should be as thoroughly performed as for the cribriform plate lymphatic vessels. It is surprising and interesting that the inflammation-

induced neo-lymphangiogenesis is specific to the cribriform plate.

We agree with the reviewer as the fact that lymphangiogenesis is unique to the cribriform plate is interesting. In fact, preliminary data from an independent group confirmed that meningeal lymphatics in both the brain and spinal cord do not change during EAE (see reference #32, Louveau, A., et al., *Nat Neurosci*, 2018). As requested by the reviewer, we analyzed lymphatic vessels in the dura in much greater detail and generated a new figure which has been included in the main body of the paper as **Revised Figure 5** and as shown below.

A new paragraph has also been included to describe these findings in the results section of the text (**page 10, lines 241 – 257**):

“Lymphangiogenesis is unique to the cribriform plate during EAE.

Recently, lymphatic vessels have been characterized in the dura and implicated as being critical for the regulation of neuroinflammation³². Additionally, meningeal lymphatics have the potential to undergo VEGFR3 dependent lymphangiogenesis after infusion of VEGFC into the CSF³. We and others⁴⁹ have demonstrated that VEGFC is up-regulated in the CNS during EAE, and here we show that VEGFC producing macrophages/dendritic cells accumulate near lymphatics dorsal to the cribriform plate. Therefore, we hypothesized that meningeal lymphatics may also undergo lymphangiogenesis. Surprisingly, meningeal lymphatics surrounding the brain show no evidence of lymphangiogenesis during EAE (**Figure 5A - B**), which has also been confirmed by others³². Despite the lack of lymphangiogenesis, meningeal lymphatics remained plastic as they underwent regression after VEGFR3 inhibition as previously described¹⁷ (**Figure 5A - E**). These data suggest that lymphangiogenesis is unique to vessels near the cribriform plate, and that their access to pro-lymphangiogenic factors differs from that of meningeal lymphatics during neuroinflammation. Additionally,

MAZ51 treatment during EAE also induces meningeal lymphatic regression although to a somewhat lesser extent when compared to steady-state conditions, suggesting that during neuroinflammation meningeal lymphatics may be slightly less sensitive to VEGFR3 tyrosine kinase inhibitors (**Figure 5 E**). We also could not observe any macroscopic changes to meningeal lymphatics surrounding the spinal cord^{17,32} (**Supplementary Figure 11**).”

We also find it interesting that lymphangiogenesis is localized to the cribriform plate despite other studies implicating meningeal lymphatics as the primary source for the drainage of CSF. We have addressed this in the discussion section of the text with the following paragraph (**pages 15 – 16, lines 372 - 389**):

“In this study we demonstrate that VEGFC is up-regulated within the CNS, and there is a substantial accumulation of VEGFC-producing macrophages and dendritic cells near the cribriform plate that consequently induce lymphangiogenesis during EAE. These data suggest that lymphatics near the cribriform plate facilitate the drainage of either: 1) soluble VEGFC produced within the CNS, 2) the drainage of CNS-derived VEGFC-producing immune cells, 3) VEGFC⁺ cells near the cribriform plate produce chemokines to recruit additional VEGFC producing cells, and/or 4) the drainage of CNS derived antigens which recruit peripheral VEGFC producing immune cells to the cribriform plate. Lymphangiogenesis occurring in vessels dorsal to the cribriform plate strongly implicate their role in the drainage of CNS derived antigens, proteins, and/or cells. In support of this, our data showing increased CNS derived antigen drainage correlates with lymphangiogenesis and implicates its role in the drainage of fluid during EAE, however, alternative pathways cannot be excluded. Nevertheless, preliminary data suggest that during EAE there are significant increases in CSF accumulation and clearance near the cribriform plate, suggesting that lymphangiogenesis may contribute to the drainage of excess fluid. Interestingly, lymphangiogenesis was not observed in meningeal lymphatics during EAE despite their implication in regulating EAE³². Lymphatic vessels on the CNS side of the cribriform plate may have easier access to CNS derived antigens, proteins, and/or cells due to the arachnoid barrier separating meningeal lymphatics from the CSF and the lack of an arachnoid barrier separating cribriform plate lymphatics from the CNS^{62,66}. Future studies are needed to elucidate the relative access of CNS derived antigens within the different lymphatics surrounding the CNS.”

Unfortunately, more thorough microscopic experiments analyzing spinal cord lymphatics is difficult as the spinal cord is relatively large and its lymphatic network heterogenous, making it difficult to quantify any changes across animals. Therefore, while data describing the meningeal lymphatics have been incorporated into the main text as **Figure 5 (see above)**, the spinal cord figure remains as supplementary data as seen in **Supplementary Figure 11**.

Reviewer 1, Major Concern #4: In Figure 2, it is not clear that the Ki67 is co-localizing with the lymphatic vessels. Figure 2T looks very similar to Figure 3S; it might be that the Ki67+ cells are proliferating immune cells in the vessels rather than the vessels themselves. Performing FACS analysis on the lymphatic endothelial cells using the Prox1-tdTomato mouse would be a better way to quantify markers of proliferation.

We thank the reviewer for noticing that Figure 2T looks similar to Figure 3S, and that the Ki67⁺ signal seen in Figure 2 may in fact be immune cells. Unfortunately, FACS analysis of lymphatic endothelial cells near the cribriform plate is quite challenging at this time due to their anatomical location, enclosure in bone, as well as their relative rarity in cell number. Nevertheless, this should be attempted in the future by pooling a large number of mice and generating novel methodology for isolating cribriform plate lymphatic endothelial cells.

In order to investigate whether our quantitation reflects Ki67⁺ immune cells or whether they are actually Ki67⁺ lymphatic endothelial cells, we generated orthogonal views of each image and re-quantified cells that are Ki67⁺ and Lyve-1⁺ in both the YZ and XZ plane, as two cells cannot occupy the same space in 3D. All cells from the original images were re-quantified using orthogonal views, and nearly all of the quantified cells were indeed Ki67⁺ lymphatic endothelial cells which resulted in nearly identical quantitation. We have included representative orthogonal views of the Ki67⁺ sections in **Figure 2** as a supplementary figure in the revised manuscript (**See Supplementary Figure 6, shown below**).

Lastly, we have also included 3D reconstruction images demonstrating the extent of Ki67⁺ proliferation of both the lymphatic endothelial cells and surrounding cells in both healthy and EAE (See **3D reconstruction videos in supplementary Figures 4 and 5**).

Reviewer 1, Major Concern #5: The authors' claims regarding CCL21/CCR7 should either be entirely removed, proven, or their language should be changed in every instance describing this data. The data presented in Figure 5 do not lead to the conclusion stated in the title of the figure.

We agree with the reviewer, and have moved the CCL21/CCR7 figure into the supplementary data as **Supplementary Figure 2**. We have also modified the title of the figure to: "CCL21 is expressed by cribriform plate lymphatics and correlates with CCR7⁺ CD11c-eYFP⁺ cell accumulation during EAE." We also included an additional citation that functionally demonstrates CCR7 is required for both dendritic cell (page 11, line 263) (See reference #50; Clarkson, B.D., et al., *Sci Rep*, 2017) and T cell (page 11, line 263) (See reference #32; Louveau, A., et al., *Nat Neurosci*, 2018) emigration from the CNS. We have also significantly changed the text throughout the paper concerning this figure to reflect that CCL21 is expressed by cribriform plate lymphatics, is up-regulated within the CNS along with CCR7, and correlates with CD11c-eYFP⁺ CCR7⁺ cells (pages 10 – 11, lines 259 – 274):

"Cribriform plate lymphatics express CCL21 and correlate with CCR7⁺ CD11c-eYFP⁺ cell accumulation.

Next, we looked to identify a potential mechanism of how CD11c⁺ cells migrate from the CNS parenchyma into lymphatic vessels near the cribriform plate. We have previously shown that the migratory chemokine receptor CCR7 is required for immune cell emigration from the CNS to the draining lymph nodes during EAE^{32,50}, similar to how T lymphocytes exit peripheral tissues⁵¹. CCL21, the ligand for CCR7, is expressed by lymphatics and is hypothesized to drive the chemotaxis of immune cells into lymphatics^{33,34,35}. Anatomically, several previous studies have reported that dendritic cells that infiltrate the CNS migrate towards the ventral regions of the olfactory bulbs near the cribriform plate during EAE, suggesting that lymphatic vessels near the cribriform plate may facilitate the drainage of dendritic cells^{43,44,45}. Here, we show that CCL21 is expressed by lymphatic vessels dorsal to the cribriform plate during steady-state conditions; however, this is accompanied by relatively sparse numbers of CCR7⁺ CD11c-eYFP⁺ cells (Supplementary Figure 2A - E). During EAE, lymphangiogenic vessels near the cribriform plate continue to express CCL21 and are accompanied by increased numbers of CCR7⁺ CD11c-eYFP⁺ cells (Supplementary Figure 2F - J). Additionally, both CCR7 and CCL21 are up-regulated within the CNS during EAE (Supplementary Figure 2K - N). Taken together, these data correlate CCR7⁺ CD11c-eYFP⁺ cell accumulation with CCL21 expression by lymphangiogenic vessels near the cribriform plate."

Reviewer 1, Major Concern #6: MAZ51 needs to be administered to healthy mice to determine whether VEGFR3 is required for maintenance – if the drug is getting rid of existing vessels, this would affect the presented analysis.

We agree with the reviewer, and have performed this experiment and included our findings in the revised manuscript as **Figure 4 (Revised Figure 4A, B, C, and D are shown below, and are part of a larger Revised Figure 4)**. Our data show that MAZ51 did not change basal levels of existing vessels near the cribriform plate. This seems to be consistent with a previous report showing that only the meningeal lymphatics of the brain and to a limited extent those surrounding the spinal cord are susceptible to regression after VEGFR3 inhibition (See reference #17; Antila, S., et al., *J Exp Med*, 2017).

We have appropriately modified the text to highlight these findings in the results and discussion

Results Section: (page 9, lines 220 – 221)

“MAZ51 also had no effect in baseline levels of cribriform plate lymphatics in vehicle treated healthy mice (Figure 4C).”

Discussion Section: (page 18, lines 444 - 450)

“Despite this, we showed that MAZ51 has no effect on lymphatic vasculature near the cribriform plate during steady-state conditions. This is consistent with previous reports in which only dural lymphatics of the brain and to a limited extent those surrounding the spinal cord were susceptible to regression after VEGFR3 inhibition¹⁷. Further studies would be needed to confirm the diffusion of MAZ51 and other VEGFR3 inhibitors *in vivo* in order to evaluate the therapeutic potential of lymphoid-modifying treatments in CNS autoimmune diseases.”

Reviewer 1, Major Concern #7: The effect of MAZ51 on other lymphatic vessels (such as the dura presented in Supplementary Figure 3) needs to be observed and quantified. The phenotype in the EAE experiments using this drug may be due to an effect on all lymphatics.

We agree with the reviewer and performed and included this experiment in the revised manuscript as **Revised Figure 5** (see response to Reviewer 1, Major Concerns #3). In summary, MAZ51 does in fact induce regression of meningeal lymphatics during EAE in addition to its role in inhibiting lymphangiogenesis in lymphatic vessels near the cribriform plate. We have appropriately modified the text to highlight the fact that lymphangiogenesis is unique to the cribriform plate in the results section that includes the new data under the section “Lymphangiogenesis is unique to the cribriform plate during EAE” (See Response to Reviewer 1, Major Concern #3; page 10, lines 241 – 257).

We also revised both the results and discussion section accordingly to reflect that MAZ51 treatment may in fact be due to its effect in other lymphatics:

Results Section: (page 10, lines 249 – 250)

“Despite the lack of lymphangiogenesis, meningeal lymphatics remained plastic as they underwent regression after VEGFR3 inhibition as previously described¹⁷ (Figure 5A – E).”

Results Section: (page 13, lines 316 – 322)

“It is important to note however that meningeal lymphatics may also play a role in regulating CNS derived antigen drainage as MAZ51 also induces meningeal lymphatic regression in both steady-state and EAE (Figure 5). This has recently been confirmed independently by another group where ablation of specifically meningeal lymphatic vessels can also reduce CSF derived antigen drainage³². Nevertheless, the uniqueness of lymphangiogenesis to the cribriform plate during EAE implicates these vessels as a distinct site for the drainage of CNS derived molecules. Therefore, both cribriform plate lymphatics and meningeal lymphatics may contribute to the drainage of CNS derived antigens³² and fluid.”

Discussion Section: (page 18, lines 443 – 450)

“MAZ51 also causes meningeal lymphatic regression during EAE, supporting that meningeal lymphatics may also play a role in the drainage of CNS derived antigens³². Despite this, we showed that MAZ51 has no effect on lymphatic vasculature near the cribriform plate during steady-state conditions. This is consistent with previous reports in which only dural lymphatics of the brain and to a limited extent those surrounding the spinal cord were susceptible to regression after VEGFR3 inhibition¹⁷. Further studies would be needed to confirm the diffusion of MAZ51 and other VEGFR3 inhibitors in vivo in order to evaluate the therapeutic potential of lymphoid-modifying treatments in CNS autoimmune diseases.”

Minor Concerns:

Reviewer 1, Minor Concern #1: Figure 2A and B are too dim and do not show the extent of proliferation of lymphatic endothelial cells as convincingly as in Figure 3.

We agree with the reviewer, as highlighting these lymphangiogenic vessels is critical for the paper. We have separated the Lyve-1 channel from the CD11c-eYFP/DAPI channels and have brightened the images in the revised **Figure 2 (Figure 2A and 2B; see response to Reviewer 1, Major Concerns #2).**

Reviewer 1, Minor Concern #2: The authors state that they cannot determine the identity of these cells within the lymphatic vessels. They need to further discuss why this is the case and what negative results they have gotten. Their conclusion that they are not neurons or other CNS-resident cells is not necessarily valid – for example, the cells might be neurons killed by the photoconversion. Furthermore, their conclusion that the new vessels are functional does not follow from the data presented – there is no evidence that the vessels transporting cells and CSF are the new lymphatic vessels rather than the existing ones.

We did not intend to imply that we have any negative results on not being able to determine the identity of these cells. We agree with the reviewer that the text was somewhat misleading and implied that photoconverted cells in the brain cannot be CNS-resident cells such as neurons, as even dead cells may still be photoconverted. Cells within the brain parenchyma as shown in **Figure 2T – W** may in fact be a CNS-resident cell, either live or dead after the photoconversion. However, we believe that any cell that has migrated out of the CNS parenchyma as shown in **Figure 2X – Eb** cannot be any CNS-resident cell such as neurons, as no CNS-resident cell can migrate outside of the CNS parenchyma. It is possible that photoconverted cellular debris from CNS-resident dead cells have drained from the CNS and are picked up outside of the CNS parenchyma within cribriform plate lymphatics, however, the images shown in **Figure 3X – Eb** show more of a cellular signal as opposed to cellular debris, either free floating or intracellular. We have modified the text to clarify this in the results section (page 8, lines 191 – 194):

“While we cannot determine the identity of these cells without additional immunolabeling, photoconverted cells

found outside of the CNS parenchyma and within cribriform plate lymphatics should exclude CNS-resident neurons, oligodendrocytes, endothelial cells, and glial cells including microglia and astrocytes.”

We also agree with the reviewer’s second point that our text is misleading and may generate confusion with the data presented; we observed and define lymphangiogenesis as both the widening of pre-existing lymphatic vessels as well as expansion of these pre-existing lymphatic vessels dorsally and laterally. In other words, the pre-existing vessel in its entirety has undergone growth and expansion instead of the generation of novel lymphatic vasculature from blood endothelial progenitor cells or infiltrating macrophages as seen in during development or lymphangiogenesis in peripheral organs. Wherever we observe immune cell and CSF accumulation reflects lymphangiogenic vessels since the vessels as a whole have grown in diameter, and have clarified this in the results section of the text (**page 6, lines 142 – 145**):

“More specifically, we observed both an expansion of pre-existing lymphatic vessels and growth of lymphatic vessels both dorsally and laterally. Lymphangiogenesis seems to be localized to the CNS side of the cribriform plate. These observations can alternatively be visualized through higher magnification or 3D reconstruction (**Supplementary Figure 3, 4, 5**).”

Reviewer 1, Minor Concern #3: In their discussion of the data in Figure 4, the authors should mention the many VEGFC+ cells that are not CD11b+.

We agree with the reviewer that there are several VEGFC+ cells that are not CD11b, suggesting that there may be other cellular sources of VEGFC during EAE. We have addressed this in the results section (**pages 9 – 10, lines 233 – 235**):

“While the majority of VEGFC producing cells are CD11b⁺ monocytes, several CD11b⁻ cells also express VEGFC suggesting that there are potentially other sources of VEGFC near the cribriform plate during neuroinflammation.”

Reviewer 1, Minor Concern #4: The authors need to quantify and discuss the clear increase in non-proliferating cells in MAZ51-treated OT-I T cells in Figure 6.

We agree with the reviewer that there seems to be an increase in non-proliferating cells in MAZ51-treated OT-I T cells in Figure 6. More specifically, there is roughly a 10% increase in non-proliferating cells in MAZ-51 treated OT-I T cells and is also reflected in the quantitation. While the histograms accurately represent this change in the quantitation, it is statistically non-significant when quantified. Nevertheless, a slight increasing trend in non-proliferating cells in the MAZ51 treated OT-I T cells supports our hypothesis that VEGFR3 dependent lymphangiogenesis may be important for the drainage of CNS derived antigens. We have included a sentence discussing this trend in the results section (**page 12, lines 303 – 306**):

“Although non-significant, there is approximately a 10% reduction in OT-I T cell proliferation in the draining lymph nodes of MAZ51 treated mice compared to vehicle treated controls, suggesting that lymphangiogenesis may partially correlate with reduced CD8 T cell proliferation (**Figure 6B, C**).”

Reviewer 1, Minor Concern #5: It would be very interesting to see an extension of the EAE clinical score curve presented in Figure 7. While it is evident that the treatment delays onset of disease, it would be interesting to see whether the MAZ51-treated mice eventually catch up to the control mice in disease severity or whether the curve peaks at a lower value.

We agree with the reviewer and have performed new experiments with an extended EAE clinical score curve (one-week extension). Our data show that the EAE clinical scores plateau after Day 18 in the MAZ51 treated group, and remain lower than vehicle treated animals. This data indicates that EAE clinical scores do not catch up to the control mice in disease severity and the curve peaks at a lower value in MAZ51 treated mice. We have included this data as **Revised Supplementary Figure 13 (Revised Supplementary Figure 13A, B, and C are shown below, and are part of a larger Revised Supplementary Figure 13)**.

Typos/Mistakes

1. On page 5, “Vascular Endothelial Growth Factor 3” is missing the word “Receptor”
2. In Figure 2 C, D, F, and G, the image labels are mixed up. - corrected
3. In Figure 3, the images say “C11” instead of “CD11” - corrected

Reviewer #2 (Remarks to the Author):

The manuscript # NCOMMS-18-16010 addresses an important question in the field of neuroimmunology regarding the mechanisms of immune cell trafficking into and outside the CNS in neuroinflammation. The authors have previously shown that dendritic cells use the CCL21/CCR7-mediated signaling pathway to traffic outside the CNS in the EAE model of human MS. In this study, they focus on a group of lymphatic vessels surrounding the olfactory mucosa and cribriform plate and demonstrate that these lymphatics become hyperproliferative at EAE score 3, respond to VEGFC which is secreted in the nose region. Moreover, blockade of VEGFC/VEGFR3 signaling using a pharmacological inhibitor MAZ-51 reduced lymphatics, disease severity after the adoptive EAE transfer and results in decreased immune cell infiltration (dendritic cells and T cells) into the CNS.

This study is highly significant since it is unclear how the lymphatics surrounding the CNS can contribute to disease pathogenesis. Although VEGFC/VEGFR3 signaling has been shown to contribute to the development and maintenance of the CNS lymphatics, their role in CNS disease pathogenesis and neuroinflammation is unclear. The study is very rigorous and the quality of the data is very high. Nevertheless, there are a few concerns that the authors need to address to consolidate some of their findings:

Reviewer 2, Major Concern #1: The authors show by immunofluorescence and Western blotting that both VEGFC and CCL21 are upregulated in the olfactory mucosa in close proximity to the position of the lymphatics (Figs. 4 and 5). However, it is unclear which cells are secreting these factors based on the images since these are secreted factors. The authors should resolve this issue by performing in situ hybridizations with probes for VEGFC and CCL21.

We agree with the reviewer that it is difficult to determine which cells are secreting VEGFC and CCL21 near the cribriform plate lymphatics as they are both soluble factors. Unfortunately, in situ hybridizations technically cannot be done for VEGFC and CCL21 as the tissues must undergo a week of decalcification in EDTA, which requires PFA fixation to prevent any degradation of antigens for immunofluorescence. Nevertheless, we hypothesize that the extensive washing in PFA and EDTA may remove any unbound VEGFC/CCL21, and the VEGFC/CCL21 signal identified by immunofluorescence should be on either the cells that express them or on the cells that express their receptors with bound VEGFC/CCL21. Our data is also supported by other studies showing VEGFC production by monocytes (as reviewed by Kerjaschki et al. JCI; reference #41), and CCL21 primarily being expressed by lymphatic endothelial cells (as demonstrated by Weber et al. Science; reference #35). We have also cited references #32 - 35 showing similar data of CCL21 expression by lymphatic endothelial cells.

Reviewer 2, Major Concern #2: When does VEGFC gets induced during disease progression to promote lymphatic proliferation? The time course of VEGFC upregulation (mRNA and protein) needs to be better described in relation to disease progression (Score 1-3) using both qRT-PCR and western blotting.

We agree with the reviewer that the temporal expression of VEGFC/VEGFR3 over the course of EAE is important as this may give insight into the progression of lymphangiogenesis. VEGFC upregulation in relation to disease progression (highly up-regulated at peak disease, with lower levels yet still higher than baseline levels during the chronic phase) has been described previously by qRT-PCR (shown by Park et al. J Histochem Cytochem; reference #49). Additionally, our data suggests that lymphangiogenesis occurs by Day 18 post-immunization, and preliminary data generated by an independent group at the University of Virginia suggests that lymphangiogenesis doesn't begin until day 13 post-immunization (shown in supplementary data by Antoine Louveau et al. Nature Neuroscience 2018; reference #32). Therefore, it seems as if lymphangiogenesis occurs between Day 13 and Day 18, which strongly correlates with disease progression and elevation of VEGFC within the CNS. More generally, levels of VEGF have been shown in both MS tissue and EAE in situ and correlated with disease progression as previously described by Martin A. Proescholdt et al. J.

Neuropathol. Exp. Neurol. 2002. This reference has been included in the results section of the text (see reference #26; page 3 – 4, lines 78 – 80).

Reviewer 2, Major Concern #3: Does administration of MAZ51 influence the lymphatics in other regions of the CNS? Although these lymphatics may not change in disease progression (EAE) as the authors have nicely shown in the supplementary data, they may be affected by MAZ51 and influence immune cell trafficking into the CNS.

We agree with the reviewer and have included these data in **Revised Figure 5**, also shown below.

Similar to a previous study by Kari Alitalo's group, I.P. injection of a VEGFR3 tyrosine kinase inhibitor causes meningeal lymphatic regression during steady-state conditions in addition to inhibiting lymphangiogenesis (see reference #17; Antila, S., et al., J Exp Med, 2017). Therefore, it is possible that the reduction in EAE severity and CNS antigen drainage may be due to both an inhibition of meningeal lymphatics and lymphangiogenesis. However, the accumulation of VEGFC/VEGFC producing cells to locally induce lymphangiogenesis is unique to cribriform plate lymphatics and does not occur in dural lymphatics, suggesting that lymphatics in the cribriform plate is a unique site for the collection of CNS derived molecules/cells. Following the reviewer's suggestion, this has been addressed in the revised in both the results and discussion section:

Results Section: (page 10, lines 249 – 250)

“Despite the lack of lymphangiogenesis, meningeal lymphatics remained plastic as they underwent regression after VEGFR3 inhibition as previously described¹⁷ (Figure 5A – E).”

Results Section: (page 13, lines 316 – 322)

“It is important to note however that meningeal lymphatics may also play a role in regulating CNS derived antigen drainage as MAZ51 also induces meningeal lymphatic regression in both steady-state and EAE (Figure 5). This has recently been confirmed independently by another group where ablation of specifically meningeal lymphatic vessels can also reduce CSF derived antigen drainage³². Nevertheless, the uniqueness of lymphangiogenesis to the cribriform plate during EAE implicates these vessels as a distinct site for the drainage of CNS derived molecules. Therefore, both cribriform plate lymphatics and meningeal lymphatics may contribute to the drainage of CNS derived antigens³² and fluid.”

Discussion Section: (page 15, lines 372 – 389)

“In this study we demonstrate that VEGFC is up-regulated within the CNS, and there is a substantial accumulation of VEGFC-producing macrophages and dendritic cells near the cribriform plate that consequently induce lymphangiogenesis during EAE. These data suggest that lymphatics near the cribriform plate facilitate the drainage of either: 1) soluble VEGFC produced within the CNS, 2) the drainage of CNS-derived VEGFC-producing immune cells, 3) VEGFC⁺ cells near the cribriform plate produce chemokines to recruit additional VEGFC producing cells, and/or 4) the drainage of CNS derived antigens which recruit peripheral VEGFC producing immune cells to the cribriform plate. Lymphangiogenesis occurring in vessels dorsal to the cribriform plate strongly implicate their role in the drainage of CNS derived antigens, proteins, and/or cells. In support of this, our data showing increased CNS derived antigen drainage correlates with lymphangiogenesis and implicates its role in the drainage of fluid during EAE, however, alternative pathways cannot be excluded. Nevertheless, preliminary data suggest that during EAE there are significant increases in CSF accumulation and clearance near the cribriform plate, suggesting that lymphangiogenesis may contribute to the drainage of excess fluid. Interestingly, lymphangiogenesis was not observed in meningeal lymphatics during EAE despite their implication in regulating EAE³². Lymphatic vessels on the CNS side of the cribriform plate may have easier access to CNS derived antigens, proteins, and/or cells due to the arachnoid barrier separating meningeal lymphatics from the CSF and the lack of an arachnoid barrier separating cribriform plate lymphatics from the CNS^{62,66}. Future studies are needed to elucidate the relative access of CNS derived antigens within the different lymphatics surrounding the CNS.”

Discussion Section: (page 18, lines 443 – 450)

“MAZ51 also causes meningeal lymphatic regression during EAE, supporting that meningeal lymphatics may also play a role in the drainage of CNS derived antigens³². Despite this, we showed that MAZ51 has no effect on lymphatic vasculature near the cribriform plate during steady-state conditions. This is consistent with previous reports in which only dural lymphatics of the brain and to a limited extent those surrounding the spinal cord were susceptible to regression after VEGFR3 inhibition¹⁷. Further studies would be needed to confirm the diffusion of MAZ51 and other VEGFR3 inhibitors in vivo in order to evaluate the therapeutic potential of lymphoid-modifying treatments in CNS autoimmune diseases.”

Reviewer 2, Major Concern #4: The kinetics of Evans blue flow the cribriform plate into the nasal cavity and deep cervical lymph nodes need to be better determined in control mice, EAE score 3.0 and EAE score 3.0 mice with MAZ51. The authors should measure dynamically over time the flow of Evans blue dye to provide convincing evidence that these lymphatics increase the drainage.

We agree with the reviewer that the kinetics of CSF efflux from the CNS between healthy and EAE animals will strongly implicate lymphangiogenic vessels in fluid drainage. We have generated novel preliminary data in an attempt to shed light on this (see Figure below). In order to measure CSF efflux over time, we injected 10 µl of gadolinium at a rate of 2 µl/minute into the cisterna magna and visualized gadolinium efflux by Magnetic Resonance Imaging (MRI). Our preliminary data suggest that CSF accumulates near the base of the brain and cribriform plate, which is consistent with a previous report (see reference #47; Ma, Q., et al., Nat Commun, 2017).

Interestingly, our data also show a significant accumulation of CSF near the cribriform plate and base of the brain over the course of 55 minutes when compared to a healthy control. This accumulation of CSF may be due to either impaired CSF efflux during EAE, or an increase in fluid accumulation due to neuroinflammation; in either case, we hypothesize that lymphangiogenesis occurs as a reactionary mechanism to help facilitate the drainage of extra CSF. MRI imaging of the deep cervical lymph nodes also reveal increased gadolinium drainage, suggesting that there is in fact increased fluid accumulation due to neuroinflammation, and lymphangiogenesis may contribute to increased drainage. This preliminary data is also supported by data shown in **Revised Figure 6**, in which there is increased CNS derived antigen drainage (which is an indirect measurement of fluid drainage as CNS derived antigen drainage is mediated by fluid; (see reference: **van Zwam, M., et al., J Mol Med, 2009**) during EAE, and pharmacological inhibition of VEGFR3 reduces CNS derived antigen drainage. This reference has been included in the text (**reference # 54**) along with the appropriate text in the results section (**page 13, lines 313 – 315**):

This data is also consistent with lymphangiogenesis contributing to increased fluid drainage from the CNS during EAE, as CNS derived antigen drainage is mediated by fluid⁵⁴.

However, it is unknown whether this increase in CSF drainage to the deep cervical lymph nodes during EAE is due to specifically lymphangiogenesis, or if any of the other routes of drainage may also compensate for increased neuroinflammation induced CSF accumulation. VEGFR3 inhibition using MAZ51 will also cause meningeal lymphatic regression (see **Figure 5 and Response to Reviewer 2, Major Concern #3**), which would confound any conclusions implicating specifically lymphangiogenic vessels in the drainage of CSF. Therefore, we do not believe this data to add any further significance that **Figure 6** did not address. Consequently, we have not included the preliminary MRI data in the revised manuscript, but have included text in the results and discussion section in an attempt to address this:

Results Section: (**page 8, lines 201 – 204**)

“Additionally, magnetic resonance imaging confirmed previous reports of CSF accumulation near the base of the brain, the cribriform plate, and the deep cervical lymph nodes over time⁴⁷ (**Supplementary Figure 10**).

Preliminary data also suggest there is increased CSF accumulation near the base of the brain, the cribriform plate, and deep cervical lymph nodes during EAE (**data not shown**).

Discussion Section: (page 15, lines 378 – 381)

“Lymphangiogenesis occurring in vessels dorsal to the cribriform plate strongly implicate their role in the drainage of CNS derived antigens, proteins, and/or cells. In support of this, our data showing increased CNS derived antigen drainage correlates with lymphangiogenesis and implicates its role in the drainage of fluid during EAE, however, alternative pathways cannot be excluded.”

We have included a portion of the MRI preliminary data in **Revised Supplementary Figure 10 (see below)** to better illustrate that CSF accumulates near the base of the brain, cribriform plate, and deep cervical lymph nodes during EAE. Previous data of Evans blue dye show CSF accumulation within the superior sagittal sinus and superficial cervical lymph nodes, which is not representative of how the majority of CSF flows and exits the CNS as pointed out by Reviewer #3 (see response to Reviewer 3, Major Concern #3).

Reviewer 2, Major Concern #5: The authors show in Figure 3 that some photoconverted cells exit the CNS and some remain in the parenchyma. What is the fraction of photoconverted cells that exit the CNS versus those that remain in the parenchyma? This is an important information for the study.

A relatively small population of cells within the CNS is photoconverted, and an even smaller fraction exits the CNS. Visualizing photoconverted cell drainage from the CNS parenchyma outside of the CNS is quite rare, with approximately 3-8 cells per animal in the lymphatic vessels near the cribriform plate. We have included a composite image of the brain to illustrate the approximate area in which cells have been photoconverted at the photoconversion site (approximately 7% of photoconverted area per brain), which reflects a mixture of both CNS resident cells and migrating immune cells. Additionally, flow cytometry analysis of brain cells photoconverted ex vivo reveal approximately 18% of cells are photoconverted after two days and PFA fixation as shown in the figure below.

We do not intend to include this data in the revised manuscript, but have clarified this in the results and methods section:

Results Section: **(page 7 – 8, lines 181 – 183)**

“This causes photoconversion of approximately 7% of the brain by area per section by confocal microscopy and 18% photoconversion when measured by cytofluorimetry (**data not shown**).”

Results Section: **(page 8, lines 189 – 190)**

“Images of photoconverted cells within cribriform plate lymphatics represent approximately 3 – 8 cells per animal.”

Methods Section: **(page 21, lines 531 – 533)**

“This protocol causes a localized photoconversion within the CNS parenchyma, yielding approximately 7% photoconversion of cells by area per brain section.”

Reviewer #3 (Remarks to the Author):

In the manuscript of Hsu et al, the authors have provided a characterization of cribriform plate (CP) lymphatic vessels and have demonstrated that they can expand during EAE. The authors have shown evidence of monocyte-derived cells producing VEGF-C in close proximity to LVs. An inhibitor of VEGFR3 led to a reduced expansion of LVs and a delayed time course of EAE. This is an interesting report that suggests that the more traditional view of CSF lymphatic outflow through the cribriform plate may be more important, at least for the initiation of EAE, than the recently rediscovered dural lymphatic vessels. However, there are many places where additional data may be beneficial to strengthen the conclusions of the study.

Reviewer 3, Major Concern #1: The authors conclude that dendritic cells are migrating through the CP rather than through dural LVs, but no evidence is presented showing a lack of cells within the dural LVs. In addition, the data (Fig 3O-S) that is shown purporting that cells are within CP lymphatics is dubious, as there are no orthogonal views showing that cells are truly within the vessels rather than sitting upon them. Indeed, there are many cells that are distributed in this region that are not associated with LVs. Wouldn't it be possible that cells could also exit through extravascular pathways through the plate to reach lymphatic vessels within the nasal cavity? A recent study has shown after intranasal administration, stem cells can reach the SAS through routes that are not associated with lymphatic vessels or the olfactory nerves (Galeano et al, Cell Transplantation, 2018, DOI: 10.1177/0963689718754561).

We agree with the reviewer that there are alternative pathways for cellular exit from the CNS including the nasal lymphatics. We did not intend to imply that dendritic cells only migrate through the cribriform plate or that these cribriform plate lymphatics are the only route of drainage. The reviewer is absolutely correct and we do identify dendritic cells also migrating within dural lymphatic vessels as well as extravascular pathways to reach lymphatic vessels within the nasal cavity such as the olfactory cranial nerve as seen in the figure below and as mentioned in the introduction (**page 3, lines 59 – 62**):

“Several alternative routes of drainage for CSF or immune cells from the CNS have also been proposed: 1) through the olfactory cranial nerves penetrating the cribriform plate, 2) other cranial nerves such as the optic nerve, 3) from the arachnoid villi into the venous sinuses, and through 4) interstitial fluid (ISF) within perivascular spaces, or the “glymphatic” system^{5,6,7,8,9,10.”}

We have also provided evidence of dendritic cell accumulation near dural lymphatics, the olfactory cranial nerve, and optic nerve in addition to lymphatics near the cribriform plate (**see below**):

The data above is not included into the figures of our manuscript as it does not add significantly to the main conclusion: neuroinflammation-induced lymphangiogenesis near the cribriform plate contributes to the drainage of CNS derived antigens, cells, and fluid.

We have also clarified this in the texts of the results and discussion:

Results Section (**Page 13, lines 316 – 319**):

“It is important to note however that meningeal lymphatics may also play a role in regulating CNS derived antigen drainage as MAZ51 also induces meningeal lymphatic regression in both steady-state and EAE (**Figure 5**). This has recently been confirmed independently by another group where ablation of specifically meningeal lymphatic vessels can also reduce CSF derived antigen drainage³².”

Discussion Section (**Page 15, lines 362 – 371**):

“Several routes have been implicated in the drainage of fluid from the CNS, and the major route(s) of drainage from the CNS have been controversial^{10,13,32,61,62}. Tracers injected into the cisterna magna suggest that drainage occurs through lymphatics at the base of the brain, lymphatics near the cribriform plate and particularly near the middle nasal turbinate, and along various cranial nerves when observed macroscopically^{5,6,7,9,10,47,63,64,65,66}. Several of these macroscopic reports only attributed either minor or non-existent drainage into dural lymphatics^{7,47}. Conversely, more microscopic studies have demonstrated that CSF infusion of various dyes and macromolecules can in fact drain into dural lymphatics^{3,17,32}, yet it is unknown how dural lymphatics access the CSF through the arachnoid barrier. Functional ablation of meningeal lymphatics results in reduced drainage to the draining lymph nodes^{17,32}, and occlusion of the cribriform plate increases intracranial pressure⁷⁵, suggesting both routes may play a role in facilitating drainage from the CNS.”

Discussion Section (**Page 15, lines 379 – 381**):

“In support of this, our data showing increased CNS derived antigen drainage correlates with lymphangiogenesis and implicates its role in the drainage of fluid during EAE, however, alternative pathways cannot be excluded.”

Discussion Section (**page 18, lines 443 – 445**)

“MAZ51 also causes meningeal lymphatic regression during EAE, supporting that meningeal lymphatics may also play a role in the drainage of CNS derived antigens³².”

We also agree with the reviewer that demonstrating CD11c-eYFP⁺ cells within cribriform plate lymphatics is difficult without orthogonal views, and have included representative orthogonal views as **Supplementary Figure 9 (see below)**. Following the reviewer’s suggestion, all cells from the original images were re-quantified using orthogonal views, and nearly all of the CD11c-eYFP⁺ cells were in fact within Lyve-1⁺ lymphatic endothelial cells resulting in nearly identical quantitation as shown in **Figure 3** and described in the legend of **Figure 3 (page 37, lines 877 - 878)**:

“Orthogonal views were used for quantitations to confirm that the CD11c-eYFP⁺ cells quantified were within Lyve-1⁺ vessels”

Reviewer 3, Major Concern #2: There is a delayed onset of EAE but there appears to be no difference in progression once the disease has initiated. Could the authors speculate on why this is the case? It would be

important to assess whether treatment with MAZ51 induced a regression of either dural LVs or CP LVs. As mentioned by the authors in the discussion, Antila et al, JEM, 2017 suggest that regression of dural LVs would take place with treatment of a tyrosine kinase inhibitor (Sunitinib) or soluble VEGFR3. One would expect that MAZ51 would also induce a regression of these vessels. It is important to try to elucidate whether the delay in EAE due to MAZ51 treatment is due to a lack of expansion of CP LVs or a regression of dural LVs (or a possible combination of both).

We agree with the reviewer, and have included novel data in **Revised Figure 4 (Revised Figure 4A, B, C, D, and E are shown below, and are part of a larger Revised Figure 4)** and as **Figure 5 (see below)** to assess whether MAZ51 induced a regression of either cribriform plate or dural lymphatics respectively. Just as the reviewer implied, MAZ51 also induces a regression of meningeal lymphatics (**Figure 5**), and it is unclear whether the delay in EAE is due to inhibition of lymphangiogenesis, inhibition of meningeal lymphatic vessel function, or probably both. We speculate both meningeal lymphatics and cribriform plate lymphatics play a role in the drainage of CNS derived antigens in the initial phase of EAE, and inhibition of these pathways results in delayed EAE onset. The reviewer is also correct in that there doesn't seem to be a difference in disease progression, however, there is a significant decrease in average maximal EAE scores in the MAZ51 treated group. We have included additional data as **Supplementary Figure 13 (see below)** into the revised manuscript in which after treatment of MAZ51, the EAE clinical scores were allowed to continue until Day 25 to better visualize the reduction in EAE severity. Lastly, we have also included additional supplementary data in **Supplementary Figure 13 (see below)** into the revised manuscript that shows treatment of MAZ51 after disease onset did not modify EAE clinical scores. This suggests that VEGFR3 plays a role in the initial phases of EAE.

The role of lymphangiogenesis versus meningeal lymphatics has been addressed in both the results and discussion section of the text:

Results Section: (page 10, lines 241 – 257):

“Lymphangiogenesis is unique to the cribriform plate during EAE.

Recently, lymphatic vessels have been characterized in the dura and implicated as being critical for the management of neuroinflammation³². Additionally, meningeal lymphatics have the potential to undergo VEGFR3 dependent lymphangiogenesis after infusion of VEGFC into the CSF³. We and others⁴⁹ have demonstrated that VEGFC is up-regulated in the CNS during EAE, and here we show that VEGFC producing macrophages/dendritic cells accumulate near lymphatics dorsal to the cribriform plate. Therefore, we hypothesized that meningeal lymphatics may also undergo lymphangiogenesis. Surprisingly, meningeal lymphatics surrounding the brain show no evidence of lymphangiogenesis during EAE (Figure 5A - B), which has also been confirmed by others³². Despite the lack of lymphangiogenesis, meningeal lymphatics remained plastic as they underwent regression after VEGFR3 inhibition as previously described¹⁷ (Figure 5A - E). These data suggest that lymphangiogenesis is unique to vessels near the cribriform plate, and that their access to pro-lymphangiogenic factors differs from that of meningeal lymphatics during neuroinflammation. Additionally, MAZ51 treatment during EAE also induces meningeal lymphatic regression although to a somewhat lesser extent when compared to steady-state conditions, suggesting that during neuroinflammation meningeal lymphatics may be slightly less sensitive to VEGFR3 tyrosine kinase inhibitors (Figure 5 E). We also could not observe any macroscopic changes to meningeal lymphatics surrounding the spinal cord^{17,32} (Supplementary Figure 11).

Results Section: (page 13, lines 316 – 322):

“It is important to note however that meningeal lymphatics may also play a role in regulating CNS derived antigen drainage as MAZ51 also induces meningeal lymphatic regression in both steady-state and EAE (Figure 5). This has recently been confirmed independently by another group where ablation of specifically meningeal lymphatic vessels can also reduce CSF derived antigen drainage³². Nevertheless, the uniqueness of lymphangiogenesis to the cribriform plate during EAE implicates these vessels as a distinct site for the drainage of CNS derived molecules. Therefore, both cribriform plate lymphatics and meningeal lymphatics may contribute to the drainage of CNS derived antigens³² and fluid.”

Results Section: (pages 13 – 14, lines 334 – 337):

“However, when MAZ51 is administered after the onset of EAE, the amelioration of EAE severity is abolished (Supplementary Figure 13). Taken together, these data identify VEGFR3 as a novel factor in the initiation of EAE, which coincides with its role in lymphangiogenesis and maintenance of meningeal lymphatics.”

Discussion Section: (page 15, lines 362 – 389):

“Several routes have been implicated in the drainage of fluid from the CNS, and the major route(s) of drainage from the CNS have been controversial^{10,13,32,61,62}. Tracers injected into the cisterna magna suggest that drainage occurs through lymphatics at the base of the brain, lymphatics near the cribriform plate and particularly near the middle nasal turbinate, and along various cranial nerves when observed macroscopically^{5,6,7,9,10,47,63,64,65,66}. Several of these macroscopic reports only attributed either minor or non-existent drainage into dural lymphatics^{7,47}. Conversely, more microscopic studies have demonstrated that CSF infusion of various dyes and macromolecules can in fact drain into dural lymphatics^{3,17,32}, yet it is unknown how dural lymphatics access the CSF through the arachnoid barrier. Functional ablation of meningeal lymphatics results in reduced drainage to the draining lymph nodes^{17,32}, and occlusion of the cribriform plate increases intracranial pressure⁷⁵, suggesting both routes may play a role in facilitating drainage from the CNS.”

In this study we demonstrate that VEGFC is up-regulated within the CNS, and there is a substantial accumulation of VEGFC-producing macrophages and dendritic cells near the cribriform plate that consequently induce lymphangiogenesis during EAE. These data suggest that lymphatics near the cribriform plate facilitate the drainage of either: 1) soluble VEGFC produced within the CNS, 2) the drainage of CNS-derived VEGFC-producing immune cells, 3) VEGFC⁺ cells near the cribriform plate produce chemokines to recruit additional VEGFC producing cells, and/or 4) the drainage of CNS derived antigens which recruit peripheral VEGFC producing immune cells to the cribriform plate. Lymphangiogenesis occurring in vessels dorsal to the cribriform plate strongly implicate their role in the drainage of CNS derived antigens, proteins, and/or cells. In support of this, our data showing increased CNS derived antigen drainage correlates with lymphangiogenesis and implicates its role in the drainage of fluid during EAE, however, alternative pathways cannot be excluded. Nevertheless, preliminary data suggest that during EAE there are significant increases in CSF accumulation and clearance near the cribriform plate, suggesting that lymphangiogenesis may contribute to the drainage of excess fluid. Interestingly, lymphangiogenesis was not observed in meningeal lymphatics during EAE despite their implication in regulating EAE³². Lymphatic vessels on the CNS side of the cribriform plate may have easier access to CNS derived antigens, proteins, and/or cells due to the arachnoid barrier separating meningeal lymphatics from the CSF and the lack of an arachnoid barrier separating cribriform plate lymphatics from the CNS^{62,66}. Future studies are needed to elucidate the relative access of CNS derived antigens within the different lymphatics surrounding the CNS.”

Reviewer 3, Major Concern #3: The functional data presented is not strong. 10 uL of 10% Evans Blue is not an appropriate tracer administration to assess CSF outflow. For one, murine CSF volume is estimated to be around 35 uL, so 10 uL is too high of a volume to administer. In addition, Evans Blue at such a high concentration likely has toxic effects in the CNS and an uncertain percentage will also bind to tissues and proteins within the SAS. A fluorescently labeled dextran (or better yet albumin) would be a more suitable tracer to use. The images presented in Supp Fig 5 do not support the claims of the authors that flow has occurred “along the superior sagittal sinus, surrounding the olfactory bulbs and flowing into the nasal cavity” One would expect that the bulk of the flow

towards the CP would occur in the SAS on the ventral side of the brain rather than at the dorsal side near the SSS (for visualization see Pizzo et al, *Physiology*, 2018, DOI: 10.1113/JP275105). Also, the image shown in Supp Fig 5 of the LNs appears to show only superficial cervical LNs, not deep cervical LNs. According to Ma et al, *Nat Comm*, 2017 (DOI: 10.1038/s41467-017-01484-6), the lymphatic route to the superficial cervical LNs (termed mandibular LNs by these authors) would occur due to outflow along optic nerves. The large amount of EB around the eyes apparent in the image in Supp Fig 5 would be supportive of this outflow pathway also playing a role.

We apologize to the reviewer for any misleading information in the text. 10 uL of Evans Blue (EB) was injected into the cisterna magna over the course of 5 minutes for a rate of 2 uL/minute. This information is clarified explained in revised **Supplementary Figure 1 (Revised Supplementary Figure 1C is shown below, and is part of a larger Revised Supplementary Figure 1)** and explored in more detail methods section (page 21, lines 535 – 538):

“Evans Blue dye injections. For experiments showing Evans Blue dye can drain from the CSF into cribriform plate lymphatics, 10 μ l of 10% Evans Blue dye was injected into the cisterna magna at a rate of 2 μ l/minute. After allowing 30 minutes for the dye to circulate, the mice were euthanized and the whole heads analyzed for Evans Blue dye distribution by microscopy (Figure 3).”

It is important to note that 10 uL of tracer injection into the cisterna magna has been published previously, and the rate of 2 uL/minute is commonly used to determine CSF efflux routes (see reference Kress B.T. et al. *Ann Neurol*. 2013) as this rate maintains physiological direction of CSF bulk flow (see references Iliff JJ et al. *Sci Transl Med*. 2012; Xie L et al. *Science*. 2013; and Iliff JJ et al. *J Neurosci*. 2013).

References:

- Kress, B.T., et al., *Impairment of paravascular clearance pathways in the aging brain*. *Ann Neurol*, 2014. **76**(6): p. 845-61.
Iliff, J.J., et al., *A paravascular pathway facilitates CSF flow through the brain parenchyma and the clearance of interstitial solutes, including amyloid beta*. *Sci Transl Med*, 2012. **4**(147): p. 147ra111.
Xie, L., et al., *Sleep drives metabolite clearance from the adult brain*. *Science*, 2013. **342**(6156): p. 373-7.
Iliff, J.J., et al., *Cerebral arterial pulsation drives paravascular CSF-interstitial fluid exchange in the murine brain*. *J Neurosci*, 2013. **33**(46): p. 18190-9.

The reviewer is also correct in that at high concentrations, EB may bind tissues and proteins it has come into contact with. We chose EB specifically for this property, as the tissues must undergo extensive fixation, decalcification, and washing. Any fluorescent tracer injected into the cisterna magna would wash away after decalcification. Additionally, EB dye has high affinity towards albumin, which is one of the most abundant protein in the CSF. Thus, we interpret the EB data as a readout of both unbound EB dye or EB bound to albumin.

Additionally, we removed the terminology “along the superior sagittal sinus, surrounding the olfactory bulbs and flowing into the nasal cavity” as we did not intend to imply that this is the primary pathway of bulk flow for CSF, simply that we observed this pathway from a dorsal view of the head. The reviewer is correct in that the majority of CSF actually flows towards the cribriform plate on the ventral side of the brain rather than the dorsal side along the superior sagittal sinus. We have observed a similar trend in which the majority of CSF flows towards the cribriform plate from the base of the brain after gadolinium injection into the cisterna magna and visualization by MRI as seen in the revised **Supplementary**

Figure 10, shown below. In order to avoid confusion, we have replaced the EB data in the supplementary figure with the MRI data which better demonstrates CSF efflux routes dynamically over time (**Revised Supplementary Figure 10**).

It is true that outflow to the cervical lymph nodes may occur along optic nerves (as observed by Ma, Q., et al., *Nat Commun*, 2017) and the accumulation of both EB dye and gadolinium near the optic nerves would confirm this. In fact, preliminary data have also identified CD11c-eYFP cell accumulation along the optic nerves among other pathways such as near dural lymphatics and olfactory cranial nerves as seen below (see response to Reviewer 3, Major Concern #1). We do not intend to include this data in the revised manuscript, but have included a section in the discussion to describe the other pathways of drainage including the optic nerve (see response to Reviewer 3 Major Concern #1).

Reviewer 3, Major Concern #4: In order to have a clinically meaningful result, the authors should perform an intervention study to determine how the progression of the disease is affected after it has already been initiated. A prevention study as is shown in the paper would not be possible in patients with MS. The authors mention several times that lymphangiogenesis may manage fluid accumulation during neuroinflammation. In this case, an inhibitor of lymphangiogenesis would not be beneficial. One would expect perhaps that the disease would worsen if fluid and inflammatory factors were allowed to accumulate within the CNS, similar to what occurs in models of psoriasis, rheumatoid arthritis or chronic airway inflammation that are cited by the authors.

We agree with the reviewer and have completed the requested study and included it in the revised manuscript as **Supplementary Figure 13** (see response to Reviewer 3, Major Concern #2). Specifically, we have done an intervention study in which MAZ51 was administered after the onset of EAE scores. Interestingly, there is no change in EAE severity during this study, suggesting that VEGFR3 may be critical for the initial development of EAE. It would be interesting in future studies to do the same intervention study in SJL mice which develop a relapse-remitting model of EAE and see if the same intervention treatment would affect subsequent relapses.

We also agree with the reviewer that autoimmunity is quite complicated as there should be a balance between decreasing antigen drainage but facilitating the drainage of fluid. Lymphangiogenesis does not occur until after disease onset, suggesting that lymphangiogenesis may be a reactionary response to deal with increased fluid accumulation. It is likely that meningeal lymphatics and baseline levels of cribriform plate lymphatics play a role in the initial phases of EAE by reducing antigen drainage, and that neuroinflammation regulates lymphangiogenesis to reduce fluid accumulation in the later stages of EAE. We have included a section in the discussion section to describe this (page 19, lines 477 – 479):

“Here, we propose that cribriform plate lymphatics plays both an immunological role by contributing to peripheral immune surveillance of the CNS, as well as a physiological role by undergoing lymphangiogenesis to drain excess fluid generated by neuroinflammation.”

Reviewer 3, Major Concern #5: The authors need to provide more compelling evidence showing that cells have truly migrated from the parenchyma to the CP lymphatics. How do the authors envision that this takes place? If migration truly occurs through the rostral migratory stream as suggested by the authors then fluorescently switched cells should be found along this stream leading to the olfactory bulb. How would the cells then transverse the SAS to reach the lymphatics?

We thank the reviewer and agree that how cells transverse the SAS to reach lymphatics near the cribriform plate is an unknown mechanism. Several groups including ours have independently demonstrated through a variety of methodologies that cells are able to migrate from the choroid plexus along the rostral migratory stream towards the olfactory bulbs during EAE (see references #43, 44, and 45), thus we felt it unnecessary to repeat this experiment using the KikGR system. It is also evident that there is a large accumulation of cells in the ventral parts of the olfactory bulbs during EAE, as well as cells in the subarachnoid space surrounding the lymphatic vessels of the cribriform plate. However, it is unknown how cells would exit the CNS and migrate into lymphatic vessels in the cribriform plate. Correlative data in this paper along, previous data by our group, as well as data generated independently by others have suggested a CCL21/CCR7 dependent mechanism driving chemotaxis of CCR7⁺ immune cells from the CNS towards lymphatics (see references #32, 50). Additionally, CSF seems to flow towards the cribriform plate as described previously (see reference #47) and shown by MRI imaging in the revised **Supplementary Figure 10** (see response to **Reviewer 3, Major Concern #3**), and these lymphatic vessels are in a prime position to take advantage of the flow of CSF. Unfortunately, the anatomical location of cribriform plate lymphatics has made it impossible to do live imaging at the cellular level; two-photon microscopy does not contain the depth penetration needed, and to our knowledge no live cell imaging in this region has been done. We have described these potential mechanisms in the revised discussion, and addressed that future experiments are needed to explore this exciting problem (page 16, lines 406 – 409):

“Nevertheless, how cells are able to traverse from the CNS parenchyma into the subarachnoid space and lymphatics near the cribriform plate remains unsolved, potentially due to the difficulty of live cell imaging in this region. Future studies are needed to explore this issue.”

Reviewer 3, Major Concern #6: The authors attempt to make the claim, without any supporting data that monocyte derived cells may be incorporating into lymphatic vessels through trans-differentiation. This is a controversial topic. Since the original reports in 2005 and 2006, several groups have attempted to lineage trace cells that are contributing to the expansion of lymphatic vessels. The conclusion that has been reached is that monocytes are likely not a major contributor to this process. If the authors want to make this claim they will need to provide evidence supporting this.

We thank the reviewer for this point and strongly agree that the ability for monocytes to transdifferentiate into lymphatic endothelial cells is quite controversial. We did not intend to imply that this is a mechanism taking place, but rather that we are unable to exclude this as a mechanism without future fate mapping studies or lineage tracing as mentioned in the results section (pages 6 – 7, lines 152 – 159) and as the reviewer has suggested:

“These data confirm our observation of *in situ* lymphangiogenesis near the cribriform plate during EAE. Additionally, preliminary data reveal increased eYFP⁺ area within Lyve-1⁺ vessels during EAE in CD11c-eYFP transgenic reporter mice (data not shown), suggesting the possibility of trans-differentiation from CD11c⁺ cells into Lyve-1⁺ cells. Therefore, we cannot exclude the possibility of trans-differentiation of infiltrating monocytes

as a mechanism of lymphangiogenesis. Consequently, it is possible that neuroinflammation-induced lymphangiogenesis near the cribriform plate is due to both proliferation or pre-existing lymphatic endothelial cells and/or trans-differentiation of infiltrating monocytes, although future lineage tracing studies are needed to explore trans-differentiation as a potential mechanism.

We have also clarified this in the discussion (page 17, lines 427 – 433):

“While our data implicate these cells as the primary source of VEGFC to promote VEGFR3-dependent lymphangiogenesis through the proliferation of pre-existing lymphatic endothelial cells, we cannot exclude the possibility that some of these cells are able to trans-differentiate into lymphatic endothelial cells themselves (Figure 3). Unfortunately, depletion studies of CD11b⁺ cells would fail to address this question as these cells are also needed to produce VEGFC to drive VEGFR3-dependent lymphangiogenesis. Future fate mapping studies are needed to truly elucidate the role of infiltrating monocyte trans-differentiation into lymphatic endothelial cell during EAE.”

Reviewer 3, Minor Concern #1: Please use scatter plots rather than bar graphs to show results.

We thank the reviewer for this input, and have appropriately modified all the bar graphs to scatter plots in the Revised Figures 2, 3, 4, 5, 6, 7, and Revised Supplementary Figures 2, 12, 13, 14.

Reviewer 3, Minor Concern #2: Please avoid use of the prefix “neo” within the term “neo-lymphangiogenesis”. Lymphangiogenesis refers to the expansion of an existing lymphatic network, which is what is suggested in the study.

We thank the reviewer and have changed all instances of the term “neo-lymphangiogenesis” to “lymphangiogenesis”. We initially used neo-lymphangiogenesis to distinguish lymphangiogenesis driven by neuroinflammation during EAE from the lymphangiogenesis that occurs during development, which may involve two separate mechanisms.

Reviewer 3, Minor Concern #3: LVs were identified “to penetrate from the middle nasal turbinate through the cribriform plate and into the subarachnoid space” How is it shown that these vessels are in direct contact with the SAS? If perineural sheaths extend along olfactory nerves as has been described in the past, then why would there be a need for a gap in the arachnoid barrier above the plate? On the other hand, if no barrier exists at this location than cells should be able to cross the plate with ease and would not need lymphatic vessels to do so.

We agree that we have not definitively shown that these vessels are in direct contact with the subarachnoid space. Anatomically, these vessels seem to be in close contact to the subarachnoid space as seen in Supplementary Figure 3A, and less obvious in Supplementary Figure 3B when the subarachnoid space is filled with cells during neuroinflammation. Additionally, we hypothesize that the lack of an arachnoid barrier separating these vessels from the subarachnoid space allows for easier access to CSF as opposed to meningeal lymphatics (see reference #62). Following the reviewer’s suggestion, we have revised the text to no longer imply direct access since we do not definitively show this. Nevertheless, we believe it is important to mention that the lack of an arachnoid barrier in this region coupled with our data showing VEGFC producing cells uniquely accumulating near cribriform plate lymphatics imply that these vessels may have higher access to CNS derived molecules/cells. This is mentioned in the revised discussion (page 15, lines 372 – 389):

“In this study we demonstrate that VEGFC is up-regulated within the CNS, and there is a substantial accumulation of VEGFC-producing macrophages and dendritic cells near the cribriform plate that consequently induce lymphangiogenesis during EAE. These data suggest that lymphatics near the cribriform plate facilitate the drainage of either: 1) soluble VEGFC produced within the CNS, 2) the drainage of CNS-derived VEGFC-producing immune cells, 3) VEGFC⁺ cells near the cribriform plate produce chemokines to recruit additional VEGFC producing cells, and/or 4) the drainage of CNS derived antigens which recruit peripheral VEGFC producing immune cells to the cribriform plate. Lymphangiogenesis occurring in vessels dorsal to the cribriform plate strongly implicate their role in the drainage of CNS derived antigens, proteins, and/or cells. In support of this, our data showing increased CNS derived antigen drainage correlates with lymphangiogenesis and implicates

its role in the drainage of fluid during EAE, however, alternative pathways cannot be excluded. Nevertheless, preliminary data suggest that during EAE there are significant increases in CSF accumulation and clearance near the cribriform plate, suggesting that lymphangiogenesis may contribute to the drainage of excess fluid. Interestingly, lymphangiogenesis was not observed in meningeal lymphatics during EAE despite their implication in regulating EAE³². Lymphatic vessels on the CNS side of the cribriform plate may have easier access to CNS derived antigens, proteins, and/or cells due to the arachnoid barrier separating meningeal lymphatics from the CSF and the lack of an arachnoid barrier separating cribriform plate lymphatics from the CNS^{62,66}. Future studies are needed to elucidate the relative access of CNS derived antigens within the different lymphatics surrounding the CNS.”

The redundancy of perineural sheaths along the olfactory nerves and lack of an arachnoid barrier is an interesting question. In non-immune privileged organs such as the skin, there is no “barrier” yet there is an extensive network of lymphatic vessels to guide fluid, antigens, and cells to the draining lymph nodes. Although perineural sheaths exist to potentially allow fluid/cells to exit and migrate to nasal lymphatics, the location of nasal lymphatics is relatively far from the cribriform plate (see reference #32). While cells could be observed traversing through the perineural sheaths along olfactory nerves, it is unknown whether they are uptaken by cribriform plate lymphatics or nasal lymphatics. These cribriform plate lymphatics also seem to exist in a prime position to sample CSF, and we hypothesize that the lack of an arachnoid barrier allows easier access to these lymphatic vessels relative to others surrounding the CNS. Nevertheless, how cells traverse from the CNS parenchyma into cribriform plate lymphatics is an interesting question. We thank the reviewer for this interesting point, agree that future studies are needed to explore this issue, and revised our manuscript accordingly (page 16, lines 406 – 409):

“Nevertheless, how cells are able to traverse from the CNS parenchyma into the subarachnoid space and lymphatics near the cribriform plate remains unsolved, potentially due to the difficulty of live cell imaging in this region. Future studies are needed to explore this issue.”

Reviewer 3, Minor Concern #4: Supp Fig 3 shows dural LVs close to olfactory bulbs with no expansion during the progression of EAE. Wouldn't these vessels be expected to be connected to the LVs at the CP?

We hypothesize that dural lymphatic vessels above the olfactory bulbs may connect to lymphatics found near the cribriform plate, however, this has never been shown. Although we would hypothesize this to be the case, we speculate that lymphangiogenesis is unique to lymphatic vessels where elevated levels of VEGFC are present. For example, lymphangiogenesis in the liver during mycobacterium infection only occurs at sites of inflammation, despite the extensive network of lymphatic vessels throughout the liver. Our data suggests that the region dorsal to the cribriform plate is uniquely the site of accumulation of VEGFC or VEGFC producing cells, which may potentially be attributed to the lack of an arachnoid barrier. We have included a section in the discussion to describe this (page 15, lines 384 – 389):

“Interestingly, lymphangiogenesis was not observed in meningeal lymphatics during EAE despite their implication in regulating EAE³². Lymphatic vessels on the CNS side of the cribriform plate may have easier access to CNS derived antigens, proteins, and/or cells due to the arachnoid barrier separating meningeal lymphatics from the CSF and the lack of an arachnoid barrier separating cribriform plate lymphatics from the CNS^{62,66}. Future studies are needed to elucidate the relative access of CNS derived antigens within the different lymphatics surrounding the CNS.”

Reviewer 3, Minor Concern #5: Supp Fig 4 and 5 are presented as evidence that there is no lymphangiogenesis occurring in the dural or spinal LVs, however, no quantifications are made of these vessels to support this.

We agree with the reviewer, and have more carefully quantified any changes to dural lymphatics during EAE and included this data in Revised Figure 5 (see response to Reviewer 3, Major Concern #2) to confirm that there is in fact no lymphangiogenesis within meningeal lymphatics during EAE. Similar to another study (see reference #32), no lymphangiogenesis could be identified in dural lymphatics during EAE as seen below.

Reviewer 3, Minor Concern #6: The authors write in several places throughout the text that historically it is well-

established that the majority of CSF flows through the cribriform plate. This is not, at the present time, the accepted dogma in the field. The dogma has persisted that arachnoid villi or projections are the major outflow route with lymphatics playing a supporting role. Since the high-profile publications describing the meningeal lymphatic vessels, this lymphatic network has also been considered to take a major part in CSF outflow. Previous studies have quantified outflow within cervical LVs and have recovered around 30 to 50% of tracers within these vessels. No study made the claim that CSF outflow predominantly occurs through lymphatic vessels until last year (Ma et al, Nat Comm, 2017). Of note, this study found several routes for outflow in addition to the nasal route.

We thank the reviewer and agree that drainage through the cribriform plate and overall the major outflow routes from the CNS are more complex as reviewed recently (see references #61 and 62). The reviewer has also described a previous publication by Ma et al. in *Nature Communications* suggesting that they could not identify drainage of fluid through dural lymphatics, which we have also included in the text as reference #47. We have highlighted in the revised text several alternative route(s) of drainage that may also contribute to the drainage of fluid, including meningeal lymphatics and arachnoid villi for humans (page 18, lines 457 – 461):

“It is unknown what the relative route(s) of drainage are in humans, however it is hypothesized that arachnoid villi contribute to the drainage of CSF much more significantly than in humans than in mice, as mice have relatively rare, if any, arachnoid villi. Nevertheless, drainage of CSF through the cribriform plate has also been observed in humans^{71, 72, 73}, and the importance of proper drainage through the cribriform plate has been demonstrated.”

We also agree that in humans the “dogma” is that the arachnoid villi or projections are the major outflow route; however, mice have relatively rare, if any, arachnoid villi or projections suggesting an alternative route of drainage. Studies done by Miles Johnston have demonstrated that drainage through the cribriform plate is conserved amongst mammals including humans (see reference #73). Nevertheless, it is unknown what the relative contribution of each pathway is in humans as mentioned above. However, a recent study has shown that drainage through the cribriform plate is impaired in Alzheimer’s disease patients, suggesting that this route may play a role as well in addition to arachnoid granulations (see reference #74). We have highlighted the arachnoid villi as a potential route of drainage in humans in the revised discussion section as shown above.

Reviewer 3, Minor Concern #7: References 16 and 17 are not the correct ones for the statement in the Introduction that LECs proliferate and undergo VEGFR3 dependent lymphangiogenesis in the meninges.

We thank the reviewer for pointing out this discrepancy, and have corrected the references.

Reviewer 3, Minor Concern #8: Introduction: “Several neuroinflammatory diseases, including MS and EAE” – these are not different diseases, one is a model for the other.

We thank the reviewer for clarifying this, and we have made the appropriate changes.

Reviewer 3, Minor Concern #9: Results: “trauma induced by intracerebral injection” – what injection are the authors referring to?

We thank the reviewer for pointing this discrepancy out, and have clarified this in the text. We were referencing the photoconversion in the KikGR experiment in which a 405 nm wavelength fiber optic cable was funneled through a needle, which was intracerebrally inserted to photoconvert the CNS parenchyma. We have changed “trauma induced by intracerebral injection” to “trauma caused by intracerebrally funneling the fiber optic cable on two consecutive days serves advantageous in recruiting additional immune cells for the photoconversion” as seen in (page 8, lines 186 – 187).

Reviewer 3, Minor Concern #10: Discussion: “Previous murine studies have also demonstrated that the majority of CSF drainage occurs through the cribriform plate” – Refs given are from the Johnston group who worked primarily with rat and sheep.

We agree that extensive studies have also been done outside of murine animals, and have removed “murine’ from the text.

Reviewer 3, Minor Concern #11: Discussion: Several of the references given for CSF drainage through the CP in humans are not supportive of this concept. For example, reference 64 is a report on LVs in the intravertebral disc. Reference 66 discusses drainage from the nasal tissue not the CNS.

We thank the reviewer for pointing out these discrepancies, and we have removed these references from the text.

Reviewer 3, Minor Concern #12: Labels on Fig 2 should read “CD11c” rather than “C11C”

We thank the reviewer for pointing out this mistake, and have corrected the figure.

Again, we thank the Editor and the reviewers for their careful work and hope that the revised manuscript is now suitable in its form for publication in Nature Communications.

Please do not hesitate to contact us if additional information would be necessary.

Sincerely yours,

Zsuzsanna Fabry
Professor
Vice Chair for Research
Chair, GEC Cellular and Molecular Pathology Graduate Program

University of Wisconsin, Madison, WI
Department of Pathology and Laboratory
Medicine, 501 SMI University of Wisconsin,
1300 University Avenue, Madison WI 53706, USA

REVIEWERS' COMMENTS:

Reviewer #1 (Remarks to the Author):

Hsu et al. have thoroughly responded to the original critiques and have greatly improved the manuscript. However, there are a few additional issues must be addressed before publication.

Major Issue

1. The authors claims that cribriform lymphangiogenesis regulates EAE, based on their MAZ51 experiments, are too strong. The authors have shown that MAZ51 also causes regression of meningeal lymphatic vessels, and it is also possible that systemic MAZ51 can regulate lymphatics elsewhere as well. Because it is not possible to specifically ablate the cribriform plate lymphatic vessels, MAZ51-related findings cannot be definitively attributed to blocking cribriform lymphangiogenesis. For instance, the paper's abstract and some of the section headings state that inhibiting lymphangiogenesis reduces T-cell proliferation and EAE severity. While this seems likely, the experiments done do not fully prove this, and the presentation of results is therefore misleading. MAZ51 experimental results should be attributed throughout the paper to VEGFR3 tyrosine kinase inhibition, with the suggestion that the effect occurs because of lymphangiogenesis inhibition, but that effects in other regions such as on existing meningeal lymphatics is also possible.

Minor Issues

1. The authors' explanation for how the cribriform plate lymphatics change during EAE is still not clear. While the authors explain the widening and lengthening of the lymphatic vessels well in their rebuttal (top of page 9), it is still not clear in the text of the manuscript as the word "expansion" could refer to either width or length. It needs to be made more explicit in the text of the manuscript as to exactly how the lymphatics are changing.

2. The section on T-cell proliferation makes claims that seem to be a stretch based on the data collected. The authors demonstrate decreased cell proliferation and claim that this is due to a decreased antigen drainage. While it is possible that the observed decrease in proliferation is due to decreased antigen drainage, this is only one of many possibilities. As the MAZ51 is delivered systemic, it could be acting directly at the lymph nodes or at some other point of T-cell activation, such as the lung (Odoardi et al., Nature 2012). And even if the decreased proliferation is indeed due to decreased antigen drainage, the effect could be related to meningeal lymphatic regression rather than decreased cribriform plate lymphangiogenesis. The authors need to alter their claims as they do not show that the cribriform plate lymphatics regulate antigen drainage.

Reviewer #2 (Remarks to the Author):

The revised manuscript # NCOMMS-18-16010A by Hsu et. al., has been significantly improved compared to the original manuscript. The authors have addressed the majority of the concerns that I had with the original manuscript in an adequate manner in particular in the following aspects:

1) The effects of MAZ51 on lymphatics in the cribriform plate and other CNS lymphatic vessels. The authors have added an important figure (Fig. 5) that demonstrates that MAZ51 induces vessel regression of other CNS lymphatics in both the healthy CNS and during EAE.

2) The authors have done a very nice job to describe the kinetics of Evans blue flow through the cribriform plate into the nasal cavity and deep cervical lymph nodes.

However, there are some minor concerns for the manuscript:

1) In figure 2, the authors show very nice images of Ki67+ cells in lymphatic vessels in healthy and EAE. Although it is very clear that the number of Ki67+ cells is higher in EAE, the quantitation in Figure 2U shows the opposite. The authors state that Ki67+ lymphatic vessels are higher in EAE. Therefore, there must be a mistake in that graph.

2) Since the authors cannot perform in situ hybridizations for VEGF-C and VEGFR3 in the cribriform plate due to technical difficulties with fixation and decalcification, the source of VEGF-C remains unclear. The authors state in discussion that VEGF-C could be produced by brain cells or infiltrating immune cells. One way that the authors could perform this experiment (at least in the brain or olfactory bulb) would be to isolate fresh frozen brain tissue and perform the in-situ with VEGFC and cell-specific markers. This way the authors can potentially exclude if the brain cells (neurons, microglia, astrocytes or endothelial cells) produce VEGF-C. I think this is a critical and novel aspect of the study that is highly significant.

However, overall the authors have done a great amount of novel work that elucidates the role of VEGFC/VEGFR3 in lymphatics in the cribriform plate in EAE and their role in immune cell trafficking.

Reviewer #3 (Remarks to the Author):

The authors have done an excellent job addressing the reviewer comments. I only have minor comments regarding some of the conclusions made within the text.

Minor comments:

1. In reference to original Major Concern #3: The addition of dynamic MRI data has significantly increased the quality of the functional data. This has made it clear that the major CSF flow pathways are ventral to the brain rather than the originally suggested dorsal pathway along the superior sagittal sinus. However, the statements of increased CSF accumulation and clearance or increased fluid drainage during EAE (Results, page 8, lines 203-204, Results, page 13, line 314, Discussion, page 15, line 382) should be removed. For one, these are based on preliminary data that is not shown to the reader. Second, one could interpret the slower dynamics of tracer accumulation that is shown in the images provided to the reviewer as a reduction in CSF clearance during EAE, rather than an increase as interpreted by the authors. Regarding the volume injected during the functional studies, the authors are correct in stating in their response to the reviewer that 10 μ L tracer has been previously published by the Nedergaard group. However, this large volume has also been criticized previously (see Hladky and Berrand, *Fluids Barriers of the CNS*, 2014). Such a high volume injected does not necessarily "maintain the physiological direction of CSF bulk flow" and may experimentally induce tracer spread against the normal flow pathways. We recommend that the authors reduce the volumes and rates injected in future studies.

2. In reference to original Major Concern #4: In response to this reviewer's comment, the authors have undertaken a study that has demonstrated that no change in disease severity was found with MAZ51 treatment after disease initiation. This is a very interesting finding, yet, it is difficult to interpret these results as some experimental details have not been included. It is not clear when after disease onset the MAZ51 treatment was started. This is not shown in figure or discussed in methods. If the lymphangiogenesis at the cribriform plate also occurs after the disease initiation then why does the inhibitor have no effect on the disease course? Also, why does the disease initiate later in this study than in the vehicle-treated mice in Supplemental Figure 13A?

3. In reference to original Major Concern #5: The authors mention in the results (page 7, lines 175-177) and the discussion (page 16, lines 395-398) that several studies have shown that CNS derived immune cells migrate towards the cribriform plate along the rostral migratory stream. The authors then cite references 43, 44 and 45 as supporting evidence for this statement. We do not agree that this has been shown conclusively. For one, all of these studies were performed ex vivo, so there has never been any evidence presented showing that the cells were "migrating" within the brain parenchyma towards the olfactory bulb or cribriform plate (which would be technically impossible at this time to show). Second, reference 44 performed Cd11c stainings and stated that the cells within the RMS were dendritic cells. However, another paper from the McMenamin group (Dando et al, *Glia*, 2016), using Cd11c-YFP mice showed that the Cd11c cells within the parenchyma may be a subtype of microglia cell, not dendritic cells. Third, reference 45 has no mention of the rostral migratory stream. The authors should tone down these statements regarding immune cell migration from the choroid plexus within the RMS.

4. In reference to original Major Concern #6: Please remove the sentences (results, pages 6-7, lines 153-159) referring to preliminary data (data not shown) in the results suggesting the possibility of Cd11c-YFP+ cells transdifferentiating into LECS (LYVE1+ cells). How could the authors differentiate between the cells that are migrating into the vessels (shown in Figure 3 and Supp Fig 9) with cells that have supposedly transdifferentiated into LECs? This can be mentioned in the discussion as a potential speculative mechanism for the expansion of the lymphatics, but without any supporting data it should be omitted from the results.

Specific comments on the text:

Introduction, page 3, lines 60-62: change "through" cranial nerves to "along" cranial nerves. Change "from" arachnoid villi to "through" arachnoid villi. Also, there is an extra "through" on line 62.

Introduction, page 3, lines 69-70: Lymphangiogenesis is critical for tumor "development" - this is not technically true, would prefer if this were changed to tumor "spread".

Results, page 5, line 107: "hypothesized" should be changed to "demonstrated" as experimental studies have shown CSF tracers accessing these vessels.

Results, page 6, line 134: how many days after EAE induction that the mice were killed for the whole-mounts should be specified.

Results, page 7, line 164: change "CSF" to "CSF-injected tracers"

Results, page 7, line 167: Is ref 42 correct for this statement?

Results, page 8, lines 195-205: please be careful to clearly differentiate between CSF and tracers injected into CSF. For example, the tracers are accumulating at the base of the brain, which is indicating the direction of CSF flow.

Results, page 10, lines 247-248: "Meningeal lymphatics surrounding the brain" - please be more specific here as to where the lymphatics were assessed

Discussion, page 15, lines 380-381: "increased antigen drainage correlates with lymphangiogenesis" - should be restated as "is associated with" as a correlation was not shown. Also, lymphangiogenesis "at the cribriform plate" should be specified - as there is mention of "alternative pathways" later in the sentence.

Discussion, page 15-16, lines 387-388: "lack of an arachnoid barrier separating cribriform plate

lymphatics from the CNS" The correct reference should be Ref 64, not Ref 66.

Reviewer #1 (Remarks to the Author):

Hsu et al. have thoroughly responded to the original critiques and have greatly improved the manuscript. However, there are a few additional issues that must be addressed before publication.

Reviewer #1, Major Concern #1:

The authors claims that cribriform lymphangiogenesis regulates EAE, based on their MAZ51 experiments, are too strong. The authors have shown that MAZ51 also causes regression of meningeal lymphatic vessels, and it is also possible that systemic MAZ51 can regulate lymphatics elsewhere as well. Because it is not possible to specifically ablate the cribriform plate lymphatic vessels, MAZ51-related findings cannot be definitively attributed to blocking cribriform lymphangiogenesis. For instance, the paper's abstract and some of the section headings state that inhibiting lymphangiogenesis reduces T-cell proliferation and EAE severity. While this seems likely, the experiments done do not fully prove this, and the presentation of results is therefore misleading. MAZ51 experimental results should be attributed throughout the paper to VEGFR3 tyrosine kinase inhibition, with the suggestion that the effect occurs because of lymphangiogenesis inhibition, but that effects in other regions such as on existing meningeal lymphatics is also possible.

We agree with the reviewer, and have changed our language to reflect that the reduction in EAE severity and CNS derived antigen specific T cell proliferation is due to VEGFR3 inhibition rather than lymphangiogenesis. In our concluding remarks for the results, we highlight that one of the ways VEGFR3 may affect T cell priming or EAE pathology is through the inhibition of lymphangiogenesis and/or the regression of meningeal lymphatics:

Specific revisions in the results section:

Page 11, Lines 963 – 971: “It is also important to note that MAZ51 not only inhibits lymphangiogenesis, but also causes meningeal lymphatic regression. However, it is technically difficult to inhibit specifically lymphangiogenic vessels near the cribriform plate because of their anatomical location and the potential access to CNS derived molecules by other lymphatics.”

Pages 11 – 12, Lines 985 – 1039: “These data suggest that VEGFR3 inhibition decreases the amount of antigen drainage to the draining lymph nodes, potentially due to inhibition of lymphangiogenesis. This data is also consistent with lymphangiogenesis contributing to increased fluid drainage from the CNS during EAE⁵⁴. Meningeal lymphatics may also play a role in regulating CNS derived antigen drainage as MAZ51 also induces meningeal lymphatic regression in both steady-state and EAE (**Figure 5**). Nevertheless, the uniqueness of lymphangiogenesis to the cribriform plate during EAE implicates these vessels as a distinct site for the drainage of CNS derived molecules. Therefore, both cribriform plate lymphatics and meningeal lymphatics may contribute to the drainage of CNS derived antigens³² and fluid.”

Page 12, Lines 43 – 53: “The reduction in EAE severity by VEGFR3 inhibition correlates with our prior observations of VEGFR3 contributing to lymphangiogenesis (**Figure 4**), CNS antigen drainage (**Figure 6**), and regression of meningeal lymphatics (**Figure 5**), suggesting that VEGFR3 may regulate EAE severity through lymphangiogenesis and/or meningeal lymphatic regression. However, when MAZ51 is administered after the onset of EAE clinical symptoms (approximately 12 days post-immunization), the amelioration of EAE severity is abolished (**Supplementary Figure 9**).”

Minor Concerns:

Reviewer 1, Minor Concern #1: The authors' explanation for how the cribriform plate lymphatics change during EAE is still not clear. While the authors explain the widening and lengthening of the lymphatic vessels well in their rebuttal (top of page 9), it is still not clear in the text of the manuscript as the word "expansion" could refer to either width or length. It needs to be made more explicit in the text of the manuscript as to exactly how the lymphatics are changing.

We have appropriately modified the text to clarify "expansion" as both the growth of pre-existing lymphatic vessels dorsally and laterally, as well as the widening of pre-existing lymphatic vessel diameter:

Specific revisions in the results section:

Page 6, Lines 409 – 410: "We observed both the expansion and widening of pre-existing lymphatic vessels dorsally and laterally at the CNS side of the cribriform plate."

Reviewer 1, Minor Concern #2: The section on T-cell proliferation makes claims that seem to be a stretch based on the data collected. The authors demonstrate decreased cell proliferation and claim that this is due to a decreased antigen drainage. While it is possible that the observed decrease in proliferation is due to decreased antigen drainage, this is only one of many possibilities. As the MAZ51 is delivered systemic, it could be acting directly at the lymph nodes or at some other point of T-cell activation, such as the lung (Odoardi et al., Nature 2012). And even if the decreased proliferation is indeed due to decreased antigen drainage, the effect could be related to meningeal lymphatic regression rather than decreased cribriform plate lymphangiogenesis. The authors need to alter their claims as they do not show that the cribriform plate lymphatics regulate antigen drainage.

We agree with the reviewer for this point, as MAZ51 inhibits both lymphangiogenesis near the cribriform plate and meningeal lymphatic regression. We have previously demonstrated that MAZ51 does not affect antigen specific T cell proliferation (**reference #23**), and have clarified this in both the results and discussion section. We have also changed our language to state that inhibition of "VEGFR3" affects CNS antigen specific T cell proliferation instead of "lymphangiogenesis", but emphasize that this is probably due to both an inhibition of lymphangiogenesis and meningeal lymphatic vessel regression.

Specific revisions in the results section:

Page 11, Lines 966 – 971: "MAZ51 was administered beginning on day 7 post-immunization to prevent any unspecific effects in the initial priming and expansion of T cells, and has previously been demonstrated to not interfere with T cell proliferation *in vitro*¹⁸. It is also important to note that MAZ51 not only inhibits lymphangiogenesis, but also causes meningeal lymphatic regression. However, it is technically difficult to inhibit specifically lymphangiogenic vessels near the cribriform plate because of their anatomical location and the potential access to CNS derived molecules by other lymphatics."

Pages 11 – 12, Lines 985 – 1039: "These data suggest that VEGFR3 inhibition decreases the amount of antigen drainage to the draining lymph nodes, potentially due to inhibition of lymphangiogenesis. This data is also consistent with lymphangiogenesis contributing to increased fluid drainage from the CNS during EAE⁵⁴. Meningeal lymphatics may also play a role in regulating CNS derived antigen drainage as MAZ51 also induces meningeal lymphatic regression in both steady-state and EAE (**Figure 5**). Nevertheless, the uniqueness of

lymphangiogenesis to the cribriform plate during EAE implicates these vessels as a distinct site for the drainage of CNS derived molecules. Therefore, both cribriform plate lymphatics and meningeal lymphatics may contribute to the drainage of CNS derived antigens³² and fluid.”

Reviewer #2 (Remarks to the Author):

The revised manuscript # NCOMMS-18-16010A by Hsu et. al., has been significantly improved compared to the original manuscript. The authors have addressed the majority of the concerns that I had with the original manuscript in an adequate manner in particular in the following aspects:

1. The effects of MAZ51 on lymphatics in the cribriform plate and other CNS lymphatic vessels. The authors have added an important figure (Fig. 5) that demonstrates that MAZ51 induces vessel regression of other CNS lymphatics in both the healthy CNS and during EAE.
2. The authors have done a very nice job to describe the kinetics of Evans blue flow through the cribriform plate into the nasal cavity and deep cervical lymph nodes.

Reviewer 2: No Major Concerns

Minor Concerns:

Reviewer 2, Minor Concern #1: In figure 2, the authors show very nice images of Ki67+ cells in lymphatic vessels in healthy and EAE. Although it is very clear that the number of Ki67+ cells is higher in EAE, the quantitation in Figure 2U shows the opposite. The authors state that Ki67+ lymphatic vessels are higher in EAE. Therefore, there must be a mistake in that graph.

We thank the reviewer for pointing this mistake out, and have corrected this mistake.

Reviewer 2, Minor Concern #2: Since the authors cannot perform in situ hybridizations for VEGF-C and VEGFR3 in the cribriform plate due to technical difficulties with fixation and decalcification, the source of VEGF-C remains unclear. The authors state in discussion that VEGF-C could be produced by brain cells or infiltrating immune cells. One way that the authors could perform this experiment(at least in the brain or olfactory bulb) would be to isolate fresh frozen brain tissue and perform the in-situ with VEGFC and cell-specific markers. This way the authors can potentially exclude if the brain cells (neurons, microglia, astrocytes or endothelial cells) produce VEGF-C. I think this is a critical and novel aspect of the study that is highly significant. However, overall the authors have done a great amount of novel work that elucidates the role of VEGFC/VEGFR3 in lymphatics in the cribriform plate in EAE and their role in immune cell trafficking.

We agree with the reviewer that this is an important question that remains, as myeloid cells may not be the only source of VEGFC especially within the CNS. Our lab is currently focusing on optimizing a novel protocol to generate a single cell suspension of the cribriform plate, which we can then test other potential source(s) of VEGFC by flow cytometry or through single cell RNA sequencing. We also agree that in situ hybridizations of solely brain tissue for VEGFC would also shed light into which cell within the CNS parenchyma contributes to VEGFC production. The role of VEGFC/VEGFR3 in regulating the initiation of EAE should be explored in future studies.

Reviewer #3 (Remarks to the Author):

The authors have done an excellent job addressing the reviewer comments. I only have minor comments regarding some of the conclusions made within the text.

Reviewer 3: No Major Concerns

Minor Concerns:

Reviewer 3, Minor Concern #1: In reference to original Major Concern #3: The addition of dynamic MRI data has significantly increased the quality of the functional data. This has made it clear that the major CSF flow pathways are ventral to the brain rather than the originally suggested dorsal pathway along the superior sagittal sinus. However, the statements of increased CSF accumulation and clearance or increased fluid drainage during EAE (Results, page 8, lines 203-204, Results, page 13, line 314, Discussion, page 15, line 382) should be removed. For one, these are based on preliminary data that is not shown to the reader. Second, one could interpret the slower dynamics of tracer accumulation that is shown in the images provided to the reviewer as a reduction in CSF clearance during EAE, rather than an increase as interpreted by the authors. Regarding the volume injected during the functional studies, the authors are correct in stating in their response to the reviewer that 10 uL tracer has been previously published by the Nedergaard group. However, this large volume has also been criticized previously (see Hladky and Berrand, Fluids Barriers of the CNS, 2014). Such a high volume injected does not necessarily "maintain the physiological direction of CSF bulk flow" and may experimentally induce tracer spread against the normal flow pathways. We recommend that the authors reduce the volumes and rates injected in future studies.

We strongly agree with the reviewer. We have removed our statements of increased CSF accumulation and clearance during EAE, as this should be further explored and characterized in greater detail and will not be the readers. It is also true that the slower dynamics of tracer accumulation surrounding the CNS may simply be due to a reduction in CSF clearance. While the dynamics of increased tracer accumulation in the draining lymph nodes of EAE support our hypothesis, further studies with more animals are needed to assess this hypothesis. We also agree that the rate of infusion of tracer into the CSF is critical, as 10 uL is excessive when compared to the total volume of CSF in C57BL/6J mice. Future MRI studies measuring the dynamics of Gadolinium drainage will be done with much lower volumes and slower infusion rates to mimic physiology as closely as possible, especially in instances where the tracer is bright.

Reviewer 3, Minor Concern #3: In reference to original Major Concern #4: In response to this reviewer's comment, the authors have undertaken a study that has demonstrated that no change in disease severity was found with MAZ51 treatment after disease initiation. This is a very interesting finding, yet, it is difficult to interpret these results as some experimental details have not been included. It is not clear when after disease onset the MAZ51 treatment was started. This is not shown in figure or discussed in methods. If the lymphangiogenesis at the cribriform plate also occurs after the disease initiation then why does the inhibitor have no effect on the disease course? Also, why does the disease initiate later in this study than in the vehicle-treated mice in Supplemental Figure 13A?

We have clarified in the results and methodology section that MAZ51 was administered the day when the first clinical sign appeared (approximately Day 12 for this particular experiment):

Specific revisions in the results section:

Page 12, Lines 051 – 053: “However, when MAZ51 is administered after the onset of EAE clinical symptoms (approximately 12 days post-immunization), the amelioration of EAE severity is abolished (**Supplementary Figure 9**).”

Specific revisions in the methods section:

Page 23, Lines 2735 – 2737: “MAZ51 or vehicle was administered beginning either on day 7 post-immunization of EAE before the onset of EAE clinical scores, or after the first sign of EAE clinical score (approximately day 12 post-immunization).”

The difference in the onset of EAE is variable (between 8 – 12 days), probably depending on the purity of MOG peptide that changes from batch to batch. In all our EAE experiments, the same batch of MOG peptide is used to immunize mice within each experiment, with batch to batch variability from one experiment to another. It is important to note however that MAZ51 delays the onset of EAE to approximately Day 14 – 15, which is outside the normalized variability of EAE onset (8 – 12 days). The reviewer also raises an interesting question; if VEGFR3 inhibition still inhibits lymphangiogenesis and causes meningeal lymphatic regression even when administered after the onset of clinical score, why does the disease maintain its course? There are several potential explanations which we hope to explore in future studies:

- 1) Antigen drainage to the draining lymph nodes is critical for the initiation of EAE, but not necessary for maintaining the disease once it has begun. This is supported by the fact that VEGFR3 inhibition prior to EAE onset also results in a delay of clinical scores. It would be interesting to treat SJL mice immunized with PLP peptide, which develop a relapse-remitting disease, with MAZ51 to see if administration after disease onset can affect subsequent relapses.
- 2) The source of antigen drainage may differentially affect the adaptive immune response. MOG drainage from the CNS or from the immunization site may differentially shape the course of EAE by skewing it towards autoimmunity or tolerance. The context in which the antigen is presented strongly skews the adaptive immune response towards one or the other; MOG antigen drainage and presentation in the CNS occurs in a different milieu than that at the immunization site. Additionally, the amount of MOG antigen draining from the immunization site is probably much larger compared to that in the CNS.
- 3) EAE clinical scores may not fully reflect the subtler pathologies that is for example regulated by fluid management, in which lymphangiogenesis may be important. Increased intracranial pressure from excessive fluid accumulation during EAE may induce more subtle neurological disorders not seen by motor deficits. It would be interesting to elucidate the role of lymphangiogenesis in specifically fluid accumulation during disease, perhaps in a model where edema is much more prevalent.

Reviewer 3, Minor Concern #3: In reference to original Major Concern #5: The authors mention in the results (page 7, lines 175-177) and the discussion (page 16, lines 395-398) that several studies have shown that CNS derived immune cells migrate towards the cribriform plate along the rostral migratory stream. The authors then cite references 43, 44 and 45 as supporting evidence for this statement. We do not agree that this has been shown conclusively. For one, all of these studies were performed *ex vivo*, so there has never been any evidence presented showing that the cells were "migrating" within the brain parenchyma towards the olfactory bulb or cribriform plate (which would be technically impossible at this time to show). Second, reference 44 performed Cd11c stainings and stated that the cells within the RMS were dendritic cells. However, another paper from the McMenamin group (Dando et al, *Glia*, 2016), using Cd11c-YFP mice showed that the Cd11c cells within the

parenchyma may be a subtype of microglia cell, not dendritic cells. Third, reference 45 has no mention of the rostral migratory stream. The authors should tone down these statements regarding immune cell migration from the choroid plexus within the RMS.

We agree with the reviewer that this hypothesized pathway has never been conclusively shown, and have toned down all of our statements involving migration along the RMS. Nevertheless, we feel it is important that we at least briefly mention this pathway as a potential migration route that warrants further investigation:

Specific revisions in the results section:

Page 7, Lines 586 – 587: “Several groups have hypothesized that CNS derived immune cells migrate towards the cribriform plate during EAE, although this has never been conclusively shown^{43,44,45}.”

Specific revisions in the discussion section:

Page 14, Lines 1214 – 1216: “Several groups have proposed that immune cells may migrate from the choroid plexus to the ventral regions of the olfactory bulbs and into cribriform plate lymphatics through the rostral migratory stream during EAE, although this has never been definitively shown^{43,44,45}.”

Reviewer 3, Minor Concern #4: In reference to original Major Concern #6: Please remove the sentences (results, pages 6-7, lines 153-159) referring to preliminary data (data not shown) in the results suggesting the possibility of Cd11c-YFP⁺ cells transdifferentiating into LECs (LYVE1⁺ cells). How could the authors differentiate between the cells that are migrating into the vessels (shown in Figure 3 and Supp Fig 9) with cells that have supposedly transdifferentiated into LECs? This can be mentioned in the discussion as a potential speculative mechanism for the expansion of the lymphatics, but without any supporting data it should be omitted from the results.

We strongly agree with the reviewer, and have removed all instances in the results section that mention CD11c⁺ cells trans-differentiating into LECs. We have maintained the possibility of trans-differentiation as a hypothesis in the discussion section to highlight that while trans-differentiation may be controversial, we cannot exclude it as a possibility as we have not done any fate mapping studies. Additionally, depletion studies cannot be done as these myeloid cells also contribute to lymphangiogenesis by producing VEGFC:

Specific revisions in the discussion Section:

Pages 15 – 16, Lines 2367 – 2451: “Lymphangiogenesis has been characterized during peripheral infection and in tumors, in which two possible mechanisms have been proposed: 1) proliferation of pre-existing lymphatic endothelial cells^{21,23,36,37,38}, or 2) trans-differentiation of infiltrating monocytes^{39,40,41}. In this study, we show an increased percentage of Ki67⁺ lymphatic endothelial cells near the cribriform plate during EAE, suggesting proliferation of pre-existing lymphatic endothelial cells may contribute at least partly to lymphangiogenesis. Whether or not trans-differentiating monocytes can contribute to lymphangiogenesis remains unclear. Interestingly, lymphatics near the cribriform plate contain higher levels of CD11c⁺ and/or CD11b⁺ cells during EAE. While our data implicate these cells as a source of VEGFC to promote VEGFR3-dependent lymphangiogenesis, we cannot exclude the possibility that some of these cells are able to trans-differentiate into lymphatic endothelial cells. Unfortunately, depletion studies of CD11b⁺ cells would fail to address this question as these cells are also needed to produce VEGFC to drive VEGFR3-dependent lymphangiogenesis. Future fate mapping studies are needed to elucidate the role of trans-differentiation.”

Reviewer 3, Specific Comments on the Text:

1. Introduction, page 3, lines 60-62: change "through" cranial nerves to "along" cranial nerves. Change "from" arachnoid villi to "through" arachnoid villi. Also, there is an extra "through" on line 62.

We have changed and correct the text.

2. Introduction, page 3, lines 69-70: Lymphangiogenesis is critical for tumor "development" - this is not technically true, would prefer if this were changed to tumor "spread".

We agree, and have changed "development" to "spread."

3. Results, page 5, line 107: "hypothesized" should be changed to "demonstrated" as experimental studies have shown CSF tracers accessing these vessels.

We agree, and have changed "hypothesized" to "demonstrated."

4. Results, page 6, line 134: how many days after EAE induction that the mice were killed for the whole-mounts should be specified.

We have clarified this in the Results section:

"However, when MAZ51 is administered after the onset of EAE when the first sign of clinical symptoms appeared (approximately day 12 post-immunization), the amelioration of EAE severity is abolished (**Supplementary Figure 9**)."

5. Results, page 7, line 164: change "CSF" to "CSF-injected tracers"

We have corrected this in the text.

6. Results, page 7, line 167: Is ref 42 correct for this statement?

We thank the reviewer for catching this, and have changed this to the correct reference.

7. Results, page 8, lines 195-205: please be careful to clearly differentiate between CSF and tracers injected into CSF. For example, the tracers are accumulating at the base of the brain, which is indicating the direction of CSF flow.

We have corrected the text to differentiate between CSF and CSF-injected tracers to clarify the difference between the two.

8. Results, page 10, lines 247-248: "Meningeal lymphatics surrounding the brain" - please be more specific here as to where the lymphatics were assessed

We have appropriately modified the text to specify the precise anatomical location where the meningeal lymphatics were assessed:

“Surprisingly, meningeal lymphatics within the confluence of sinuses between the transverse and superior sagittal sinus show no evidence of lymphangiogenesis during EAE (**Figure 5A - B**), which has also been confirmed by others³².”

9. Discussion, page 15, lines 380-381: "increased antigen drainage correlates with lymphangiogenesis" - should be restated as "is associated with" as a correlation was not shown. Also, lymphangiogenesis "at the cribriform plate" should be specified - as there is mention of "alternative pathways" later in the sentence.

We have made the appropriate changes to the text.

10. Discussion, page 15-16, lines 387-388: "lack of an arachnoid barrier separating cribriform plate lymphatics from the CNS" The correct reference should be Ref 64, not Ref 66.

We have corrected this reference.

Again, we thank the Editor and the reviewers for their careful work and hope that the revised manuscript is now suitable in its form for publication in Nature Communications.

Please do not hesitate to contact us if additional information would be necessary.

Sincerely yours,

Zsuzsanna Fabry
Professor
Vice Chair for Research
Chair, GEC Cellular and Molecular Pathology Graduate Program

University of Wisconsin, Madison, WI
Department of Pathology and Laboratory
Medicine, 501 SMI University of Wisconsin,
1300 University Avenue, Madison WI 53706, USA